# Understanding the Kronecker Matrix-Vector Complexity of Linear Algebra

Raphael A. Meyer [1]   William Swartworth [2]   David P. Woodruff [2]

## Abstract

We study the computational model where we can access a matrix $\mathbf{A}$ only by computing matrix-vector products $\mathbf{A}x$ for vectors of the form $x = x_1 \otimes \cdots \otimes x_q$. We prove exponential lower bounds on the number of queries needed to estimate various properties, including the trace and the top eigenvalue of $\mathbf{A}$. Our proofs hold for all adaptive algorithms, modulo a mild conditioning assumption on the algorithm's queries. We further prove that algorithms whose queries come from a small alphabet (e.g., $x_i \in \{\pm 1\}^n$) cannot test if $\mathbf{A}$ is identically zero with polynomial complexity, despite the fact that a single query using Gaussian vectors solves the problem with probability 1. In steep contrast to the non-Kronecker case, this shows that sketching $\mathbf{A}$ with different distributions of the same subguassian norm can yield exponentially different query complexities. Our proofs follow from the observation that random vectors with Kronecker structure have exponentially smaller inner products than their non-Kronecker counterparts.

## 1. Introduction

Tensors have emerged as a canonical way to represent multi-modal or very high-dimensional datasets in areas ranging from quantum information science (Biamonte, 2019) to medical imaging (Selvan & Dam, 2020; Sedighin, 2024). Such applications often result in compact representations of tensors. For instance, applications in quantum information theory use the so-called PEPS network or other compact tensor networks, while applications in partial differential equations often use tucker or tensor train decompositions.

These applications overcome the curse of dimensionality by representing an underlying high dimensional linear operator as a network of a series of low dimensional tensors. Abstractly, in these applications we are given an order $2q$ tensor $\mathcal{A} \in (\mathbb{R}^n)^{\otimes 2q}$ that represents a linear operator from $(\mathbb{R}^n)^{\otimes q}$ to $(\mathbb{R}^n)^{\otimes q}$, and we often want to approximately compute some properties of this linear operator, such as its trace or spectral norm.

By appropriately reordering the entries of $\mathcal{A}$, we can explicitly write down a matrix $\mathbf{A} \in \mathbb{R}^{n^q \times n^q}$ that describes this linear operator. Our goal then becomes to estimate the trace, spectral sum, operator norm, or some other property of $\mathbf{A}$. However, since we may not know the structure of the underlying compact representation, we would like to estimate properties of $\mathbf{A}$ without explicitly forming $\mathbf{A}$, as doing so would break our compact representation of $\mathcal{A}$. Instead we take advantage of our compact representation to efficiently and implicitly access $\mathbf{A}$ through linear measurements, such as the Kronecker matrix vector product:

**Definition 1.** *Let $\mathbf{A} \in \mathbb{R}^{n^q \times n^q}$. Then Kronecker Matrix-Vector Product Oracle is an oracle that, given $\mathbf{x}_1, \ldots, \mathbf{x}_q \in \mathbb{R}^n$, returns $\mathbf{A}\mathbf{x} \in \mathbb{R}^{n^q}$ where $\mathbf{x} = \otimes_{i=1}^{q} \mathbf{x}_i$. Here, $\otimes$ denotes the Kronecker product.*

For many different compact representations of $\mathcal{A}$, it is possible to compute some compact representation of the Kronecker matrix-vector product $\mathbf{A}\mathbf{x}$ efficiently (Lee & Cichocki, 2014; Feldman et al., 2022). This is done in many algorithms and can go by different names, such as Khatri-Rao sketching or rank-one measurements. However, these algorithms tend to make strong assumptions about the structure of $\mathbf{A}$ in order to achieve a polynomial runtime (Al Daas et al., 2023; Li et al., 2017) or obtain a worst-case runtime that is exponential in $q$ (Meyer & Avron, 2023; Avron et al., 2014; Song et al., 2019b). It has been unclear whether this exponential cost is unavoidable and what structure in $\mathbf{A}$ leads to this expensive runtime. In this paper, we address this question by demonstrating explicit constructions of $\mathbf{A}$ that elicit these lower bounds.

Algorithms for fast tensor computations are well studied. There is a large number of randomized algorithms that provide strong approximation guarantees to a tensor and are very efficient. Although not all in the Kronecker matrix-vector model, many such applications involve making linear

---

[1]Computing + Mathematical Sciences Department, California Institute of Technology. [2]Department of Computer Science, Carnegie Mellon University. Correspondence to: Raphael Meyer <ram900@caltech.edu>, William Swartworth <wswartwo@andrew.cmu.edu>, David P. Woodruff <dwoodruf@cs.cmu.edu>.

*Proceedings of the $42^{nd}$ International Conference on Machine Learning*, Vancouver, Canada. PMLR 267, 2025. Copyright 2025 by the author(s).

measurements with a rank-one tensor, for which our techniques may apply. Further, there is a body of work on lower bounds for tensor algorithms. This work often focuses on complexity classes, for instance showing that computing the spectral norm of $\mathcal{A}$ is NP-Hard. However, it is not clear how this relates to the number of Kronecker matrix-vector products it takes to estimate a property of $\mathbf{A}$, which is the focus of our paper. Relatively little research focuses on query complexity lower bounds for tensor computations.

In this paper, we leverage a novel observation about the orthogonality of random Kronecker-structured vectors in order to prove exponential lower bounds on the number of Kronecker matrix-vector products needed to approximately compute properties of $\mathbf{A}$. We show that any algorithm which can estimate the trace or spectral norm of $\mathbf{A}$ to even low accuracy must use a number of Kronecker matrix-vector products that is exponential in $q$, modulo a mild assumption on the conditioning of the algorithm:

**Theorem 2** (Informal version of Theorem 6). *Any "well-conditioned" algorithm must compute $t \geq C^q$ Kronecker matrix-vector products with $\mathbf{A}$ to return an estimate $z \in (1 \pm \frac{1}{2})\lambda_1(\mathbf{A})$ with probability at least $\frac{2}{3}$.*

**Theorem 3** (Informal version of Theorem 7). *Any "well-conditioned" algorithm must compute $t \geq C^q$ Kronecker vector-matrix-vector products with $\mathbf{A}$ to return an estimate $z \in (1 \pm \frac{1}{2}) \operatorname{tr}(\mathbf{A})$ with probability at least $\frac{2}{3}$.*

This explains why methods such as Kronecker JL and Kronecker Hutchinson require exponential sketching dimension as observed by several prior works (Meyer & Avron, 2023; Ahle & Knudsen, 2019). Phrased another way, this analysis explains why the Kronecker matrix-vector complexity of linear algebra problems is exponentially higher than the classical (non-Kronecker) matrix-vector complexity.

Our core orthogonality observation also implies another gap between Kronecker and non-Kronecker matrix-vector complexities. We show that for the zero testing problem, there is an exponential gap between sketching with the Kronecker product of Gaussian vectors versus Rademacher vectors. It suffices to using a single query with the Kronecker of Gaussian vectors to test if $\mathbf{A}$ is zero, but it takes $\Theta(2^q)$ queries with Rademacher vectors.

**Theorem 4** (Special case of Theorem 18). *For any Kronecker matrix-vector algorithm whose query vectors $\mathbf{v} = \otimes_{i=1}^{q} \mathbf{v}_i$ are built from the Rademacher alphabet $\mathbf{v}_i \in \{\pm 1\}^n$, it is necessary and sufficient to use $\Theta(2^q)$ to test if $\mathbf{A} = \mathbf{0}$ or if $\mathbf{A} \neq \mathbf{0}$.*

The difference between using "small alphabets" (e.g., Rademacher vectors) and "large alphabets" (e.g., Gaussian vectors) almost never asymptotically matters in the non-Kronecker case, where we expect that all algorithms that use subgaussian variables achieve the same asymptotic

performance. In contrast, we demonstrate that *having subgaussianity does not suffice to understand the complexity of Kronecker matrix-vector algorithms*. Analogously, we show that there can also be a gap between using complex-valued and real-valued queries, which again does not typically matter in the non-Kronecker case. As a byproduct of our analysis, we prove that an algorithm of (Meyer & Avron, 2023) has a near-optimal sample complexity for trace estimation.

Broadly, our analysis reveals new insights on the fundamental complexity of linear algebra in the Kronecker matrix-vector model. We show that basic linear algebra tasks must incur an exponential sample complexity in the worst case. So, if we wish to have faster algorithms then we need to assume that $\mathbf{A}$ has some structure that avoids the worst-case structure imposed by these lower bounds. Further, we show that when designing randomized algorithms for the Kronecker matrix-vector model, it is important to examine our base random variables more closely that we may in the non-Kronecker case, as the choice of two similar variables like Rademachers and Gaussians may incur an additional exponential cost.

The rest of the paper is structured as follows: we first discuss related work. In Section 2 we introduce notation. In Section 3 we explain our theorem statements in more detail. In Section 4 we prove our lower bounds on trace estimation and spectral norm approximation against all Kronecker matrix-vector algorithms. In Section 5 we prove our lower bounds against small alphabet algorithms for the zero testing problem.

### 1.1. Related Work

Tensors have a long history of study in the sketching literature, particularly for the problem of $\ell_2$ norm estimation (Ahle et al., 2020; Ahle & Knudsen, 2019; Pham & Pagh, 2013). Previously (Ahle & Knudsen, 2019) observed that a Kronecker-structure $\ell_2$ embedding cannot work with fewer than exponential measurements in the number of modes. This is why (Ahle & Knudsen, 2019) require a more complicated sketch to construct their embeddings for high-mode tensors. However it does not appear to be known whether a Kronecker-structured sketch could work for $\ell_2$ estimation, using a subexponential number of measurements, if one drops the requirement that the sketch be an embedding. Our work partially resolves this by showing that any such sketching matrix must be extremely poorly conditioned. There is also a large body of work on sketching Tucker, tensor train, tree networks, and general tensor networks, see, e.g., (Song et al., 2019a; Mahankali et al., 2024) and the references therein.

## 2. Preliminaries

We use capital bold letters ($\mathbf{A},\mathbf{B},\mathbf{C}$,...) to denote matrices, lowercase bold letters to denote vectors ($\mathbf{a},\mathbf{b},\mathbf{c}$,...), and lowercase non-bold letters to denote scalars (a,b,c,...). $\mathbb{R}$ is the set of reals, $\mathbb{C}$ is the set of complex numbers, and $\mathbb{N}$ is the set of natural numbers. We will let $\mathcal{A}$ denote an algorithm. $\mathbf{x}^\mathsf{T}$ denotes the transpose and $\mathbf{x}^\mathsf{H}$ denotes the conjugate transpose. We use bracket notation $[\mathbf{a}]_i$ to denote the $i^{th}$ entry of $\mathbf{a}$ and $[\mathbf{A}]_{i,j}$ to denote the $(i, j)$ entry of $\mathbf{A}$. $\|\mathbf{a}\|_2$ denotes the L2 norm of a vector. $\otimes$ denotes the Kronecker product. $\mathrm{tr}(\mathbf{A})$ is the trace of a matrix. We let $[n] = \{1, \dots, n\}$ be the set of integers from 1 to $n$. For probability distributions $\mathbb{P}$ and $\mathbb{Q}$ on space $(\Omega, \mathcal{F})$, $D_{TV}(\mathbb{P}, \mathbb{Q})$ is the total variation distance between $\mathbb{P}$ and $\mathbb{Q}$, and $D_{KL}(\mathbb{P} \,\|\, \mathbb{Q})$ is the Kullback-Liebler divergence. We will let $\mathcal{L} \subseteq \mathbb{C}$ denote an alphabet.

## 3. Technical Overview

In this section, we state our core technical results more precisely and discuss their context while delaying their proof details to later in the paper. We start with the discussion of lower bounds against not ill-conditioned algorithms for trace estimation and spectral norm computation. Then, we go into the discussion of zero testing and the insufficiency of subgaussianity to understand Kronecker matrix-vector complexity.

### 3.1. Lower Bounds on Trace and Spectral Norm Estimation

In this section we formally state Theorem 2 and Theorem 3, our lower bounds against approximating the trace and the spectral norm of a matrix. Both lower bounds hold against algorithms that are not ill-conditioned. So, we first take a moment to formalize this conditioning:

**Definition 5.** *Fix a matrix-vector algorithm Algo. For any input matrix $\mathbf{A}$, let $\mathbf{v}^{(1)}, \dots, \mathbf{v}^{(t)}$ be the matrix-vector queries computed by Algo, and let $\mathbf{V} := [\mathbf{v}^{(1)} \; \cdots \; \mathbf{v}^{(t)}] \in \mathbb{R}^{n^q \times t}$ be the matrix formed by concatenating these vectors. If we know that for all inputs $\mathbf{A}$ we have that the condition number of $\mathbf{V}$ is at most $\kappa$, then we say that Algo is $\kappa-$conditioned.*

We prove lower bounds against Kronecker matrix-vector algorithms that are $\kappa-$conditioned. We will momentarily discuss how mild this conditioning assumption is. First, we state our formal results:

**Theorem 6.** *Any $\kappa-$conditioned Kronecker matrix-vector algorithm that can estimate the spectral norm of any symmetric matrix $\mathbf{A}$ to multiplicative less than error $C_\tau^{q/2}$ with probability at least $\frac{2}{3}$ must use at least $t = \Omega(\min\{C_0^{q/2}, \frac{C_\tau^{q/2}}{\kappa^2}\})$ Kronecker matrix-vector products.*

**Theorem 7.** *Any $\kappa-$conditioned Kronecker vector-matrix-vector algorithm that can estimate the trace of any PSD matrix $\mathbf{A}$ to relative error $(1 \pm \varepsilon)$ with probability at least $\frac{2}{3}$ must use at least $t = \Omega(\min\{C_0^{q/2}, \frac{C_\tau^{q/2}}{\kappa^2 \sqrt{\varepsilon}}\})$ Kronecker matrix-vector products.*

Here, $C_\tau$ and $C_0$ are constants greater than 1 which are specified in Lemma 8. Notice that the first result is against *matrix-vector* methods where we can compute $\mathbf{Ax}$, while the second is against *vector-matrix-vector* methods where we can compute $\mathbf{x}^\mathsf{T}\mathbf{Ax}$. The proofs for these results both follow from strong orthogonality between random Kronecker structured vectors. Formally, we rely on the following observation:

**Lemma 8.** *Let $\mathbf{u} = \mathbf{u}_1 \otimes \cdots \otimes \mathbf{u}_q$ where $\mathbf{u}_i$ is a uniformly random unit vector in $\mathbb{R}^n$. Then, for any $\mathbf{v} = \mathbf{v}_1 \otimes \cdots \mathbf{v}_q$ where each $\mathbf{v}_i$ is an arbitrary unit vector in $\mathbb{R}^n$, we have that*

$$\Pr\left[\langle \mathbf{u}, \mathbf{v}\rangle^2 \geq \frac{C_\tau^{-q}}{n^q}\right] \leq C_0^{-q}$$

*For some universal constants $C_\tau, C_0 > 1$.*

We prove Lemma 8 in Appendix A. What makes Lemma 8 unique is the rate of $C_\tau^{-q}$ inside the probability. This is because for a uniformly random unit vector $\mathbf{a} \in \mathbb{R}^{n^q}$ and arbitrary unit vector $\mathbf{b} \in \mathbb{R}^{n^q}$, we instead expect that $\langle \mathbf{a}, \mathbf{b}\rangle^2 \approx \frac{1}{n^q}$. So, in contrast, Lemma 8 shows an exponentially smaller inner product between random Kronecker-structured vectors. We will momentarily explain how Lemma 8 translates into the lower bounds of Theorems 6 and 7, but first we take a moment to discuss the strength of the conditioning assumption.

To understand the weight of this conditioning assumption, we take a moment to examine some of the most common Kronecker matrix-vector algorithms: Khatri-Rao Sketches. A Khatri-Rao Sketch is a non-adaptive Kronecker matrix-vector product and typically takes each query vector to be the Kronecker product of $q$ iid copies of some subgaussian vector. That is, $\mathbf{v}^{(i)} = \otimes_{j=1}^q \mathbf{v}_j^{(i)}$ where $\mathbf{v}_j^{(i)} \sim \mathcal{D}$ for some isotropic distribution $\mathcal{D}$. For instance, Kronecker JL and Kronecker Hutchinson use Khatri-Rao Sketching (Jin et al., 2021; Sun et al., 2021; Feldman et al., 2022; Bujanovic & Kressner, 2021; Meyer & Avron, 2023; Lam et al., 2024). In these cases, we should expect $\mathbf{V} = [\mathbf{v}^{(1)} \; \cdots \; \mathbf{v}^{(t)}]$ to be extremely well conditioned. This is because the inner products between the query vectors tensorizes as in Lemma 8:

$$\langle \mathbf{v}^{(1)}, \mathbf{v}^{(2)}\rangle = \prod_{i=1}^q \langle \mathbf{v}_i^{(1)}, \mathbf{v}_i^{(2)}\rangle$$

If the sketching vectors come from a continuous random distribution, then Lemma 8 tells us that these vectors have exponentially small inner product. If the sketching vectors

instead come from a discrete random distribution, then Theorem 18 in Section 5 shows that the inner product will be exactly zero with very high probability. Either way, the matrix $\mathbf{V}$ has nearly orthogonal columns with very high probability, in turn implying that the condition number of $\mathbf{V}$ is at most $O(1)$ with very high probability. So, any Khatri-Rao sketching method must incur the exponential lower bounds in Theorems 6 and 7. More broadly, we are not aware of any Kronecker matrix-vector algorithm that whose condition number is exponential in $q$, which is the degree of ill-conditioning required to avoid incurring the exponential lower bound.

### 3.2. Zero Testing

Consider the following very simple problem:

**Problem 9.** *Let* $\mathbf{A} \in \mathbb{R}^{n^q \times n^q}$ *be a matrix. Using only matrix-vector products with* $\mathbf{A}$*, decide if* $\mathbf{A} = \mathbf{0}$ *or if* $\mathbf{A} \neq \mathbf{0}$*. Be correct with probability at least* $\frac{2}{3}$*.*

When we are allowed to use classical (non-Kronecker) matrix-vector products, then we can solve Problem 9 with a single Gaussian query with probability 1. The same holds in the Kronecker matrix-vector case: if we let $\mathbf{v} = \mathbf{g}_1 \otimes \cdots \otimes \mathbf{g}_q$ where $\mathbf{g}_i \sim \mathcal{N}(\mathbf{0}, \mathbf{I})$ then $\mathbf{A}\mathbf{v} \neq \mathbf{0}$ with probability one if $\mathbf{A} \neq \mathbf{0}$. This is a direct consequence of the kernel of any nonzero matrix being a set of measure zero.

This low query complexity remains true in the classical (non-Kronecker) case if we restrict ourselves to use Rademacher vectors. More formally, if we only allow ourselves to compute $\mathbf{A}\mathbf{x}$ for vectors $\mathbf{x} \in \{\pm 1\}^{n^q}$, then using $O(1)$ matrix-vector products suffices to solve Problem 9. This follows from many possible results, including applying Hutchinson's estimator to $\mathbf{A}^\mathsf{T}\mathbf{A}$ to estimate $\|\mathbf{A}\|_F^2$ to constant factor accuracy with $O(1)$ queries (Meyer & Avron, 2023).

This story is ubiquitous in matrix-vector complexity – changing the base distribution we sample with from any subgaussian distribution to any other subgaussian distribution (e.g. from Gaussian to Rademacher) does not change this asymptotic complexity of solving linear algebra problems (Saibaba & Mikedlar, 2025; Meyer et al., 2021).

We now show that this story fails to hold true in the Kronecker matrix-vector setting:

**Theorem 10.** *Consider any Kronecker matrix-vector algorithm that only computes product using query vectors of the form* $\mathbf{v} = \mathbf{v}_1 \otimes \cdots \otimes \mathbf{v}_q$ *where* $\mathbf{v}_i \in \{\pm 1\}^n$*. Then, this algorithm needs* $\Theta(2^q)$ *queries to solve* Problem 9*.*

Not only does building queries with the Kronecker product of Rademacher entries not suffice, but no algorithm that uses $\{\pm 1\}$ entries can efficiently test if a matrix is zero. This steeply violates the story we would expect to hold from the classical (non-Kronecker) case. We prove a generalization

of this results in Theorem 18 of Section 5, where we only allow the entries of vectors to belong to a fixed alphabet $\mathcal{L} \subset \mathbb{R}$. For large enough $n$, we show that $(1 - \Theta(\frac{1}{|\mathcal{L}|}))^q$ queries are necessary and sufficient to solve the zero-testing problem. In other words, we must pay a cost that is exponential in $q$ unless $|\mathcal{L}| = \Omega(q)$. Broadly this tell us the following:

> *Knowing that a random vector is subgaussian does not suffice to tightly bound the query complexity of the Kronecker matrix-vector algorithm using that variable.*

We also note that any algorithm that can estimate the trace of a PSD matrix to relative error $O(1)$ can be used to solve Problem 9. In particular, it is worth comparing Theorem 10 to Table 1 from (Meyer & Avron, 2023). (Meyer & Avron, 2023) show an algorithm that uses the Kronecker product of Rademacher vectors to estimate the trace of a PSD matrix to constant factor error using $O(2^q)$ queries when $n = 2$. They also show that the same algorithm run with uniformly random unit vectors instead of Rademacher vectors achieves the same result in just $O(1.5^q)$ queries.

We can therefore conclude from Theorem 10 that the optimal query complexity of any algorithm that solves trace estimation while using the $\{\pm 1\}$ alphabet is therefore $\Theta(2^q)$ when $n = 2$. Since we know that $O(1.5^q)$ is possible with continuous variables, we prove for the first time that the task of *trace estimation cannot be solved with optimal query complexity using Rademacher vectors*.

Again, this reinforces how the choice of base subgaussian distribution can exponentially change our final sample complexity. The core of the proofs here also rely on orthogonality. We show that with overwhelming probability, random Kronecker-structured vectors built from a small alphabet are almost surely perfectly orthogonal:

**Lemma 11.** *There exists a distribution over random vectors* $\mathbf{u} \in \mathbb{R}^{n^q}$ *such that every fixed vector* $\mathbf{v} = \mathbf{v}_1 \otimes \cdots \otimes \mathbf{v}_q$ *with* $\mathbf{v}_i \in \{\pm 1\}^n$ *has* $\langle \mathbf{u}, \mathbf{v} \rangle = 0$ *with probability at least* $1 - \frac{1}{2^q}$*.*

Again, we prove this result in broader generality in Section 5, with respect to an arbitrary alphabet.

We also take a moment to reflect further on another observation from (Meyer & Avron, 2023): the complex Kronecker matrix-vector oracle is different from the real Kronecker matrix-vector oracle. That is, if we allow $\mathbf{v} = \otimes_{i=1}^q \mathbf{v}_i$ where $\mathbf{v}_i \in \mathbb{C}^n$, this model is more expressive than the real-valued Kronecker matrix-vector model. In particular, it takes up to $2^q$ real-valued Kronecker matrix-vector products to simulate computing a single complex Kronecker matrix-vector product. We also analyze the complex case for the zero-testing problem, and show that zero testing with the $\{\pm 1, \pm i\}$ alphabet requires $\Omega(1.25^q)$ queries, establishing

that this is easier than the zero testing in the real $\{\pm 1\}$ alphabet. However, this difference in base of exponent between $2^q$ and $1.25^q$ may also be attributed to the difference in the size of the alphabet, and so it remains unclear how to make an apples-to-apples comparison of the real and complex models and show that the complex model is fundamentally more query efficient.

# 4. Proving Theorem 6 and Theorem 7

In this section, we outline the proof techniques for Theorem 6 and Theorem 7. Both lower bounds rely on Lemma 8 as a starting point, as we can plant a very large random Kronecker-structured vector on some Gaussian data. Since the inner product between our queries $\mathbf{v} \in \mathbb{R}^{n^q}$ and the planted vector $\mathbf{u} \in \mathbb{R}^{n^q}$ is tiny, our queries cannot reliably identify the planted structure $\mathbf{u}$. More specifically, the noise from the Gaussian data hides the impact of the inner product between $\mathbf{v}$ and $\mathbf{u}$ on our queries. In the following subsections, we formalize this idea.

More broadly, our proofs hold against adaptive algorithms. That is, the algorithm is allowed to use previous responses from the oracle to decide what query to compute next. We handle adaptivity by generalizing the proof techniques in (Simchowitz et al., 2017), who propose an information-theoretic structure to lower bound the number of matrix-vector products needed to solve certain linear algebra problems. In Appendix D, we generalize their techniques in order to give lower bounds against *arbitrary constrained matrix-vector models*. For instance, while we constrain ourselves to use Kronecker-structured matrix-vector products, we could instead analyze sparse query vectors instead though this model. We leave the broader implications of this generalized lower bound as future work.

## 4.1. Proof Sketch of Trace Estimation Lower Bound

We now outline the proof of Theorem 7. We start by invoking a related but different query complexity problem in a related but different computational model.

**Definition 12.** *Fix a vector* $\mathbf{a} \in \mathbb{R}^{n^q}$. *The Kronecker-Structured Linear Measurement Oracle for* $\mathbf{a}$ *is the oracle that, given any vectors* $\mathbf{v}_1, \ldots, \mathbf{v}_q \in \mathbb{R}^n$, *returns the inner product* $\langle \mathbf{a}, \mathbf{x}_1 \otimes \cdots \otimes \mathbf{x}_q \rangle \in \mathbb{R}$.

$$(\mathbf{x}_1, \cdots, \mathbf{x}_k) \xRightarrow{input} \text{ORACLE} \xRightarrow{output} \langle \mathbf{a}, \mathbf{x}_1 \otimes \cdots \otimes \mathbf{x}_q \rangle$$

**Theorem 13.** *Any* $\kappa-$*conditioned Kronecker-Structured Linear Measurement algorithm that can estimate the squared L2 norm* $\|\mathbf{a}\|_2^2$ *to relative error* $(1 \pm \varepsilon)$ *with probability* $\frac{2}{3}$ *must use at least* $t = \Omega(\min\{C_0^{q/2}, \frac{C_\tau^{q/2}}{\kappa^2 \sqrt{\varepsilon}}\})$ *queries.*

First note that Theorem 13 suffices to prove Theorem 7. This is because any vector-matrix-vector trace estimation method

can be used to construct a linear measurement algorithm. That is, suppose that some vector-matrix-vector algorithm can estimate the trace of any PSD matrix $\mathbf{A} \in \mathbb{R}^{n^q \times n^q}$ with probability $\frac{2}{3}$ using $t$ queries. We can then fix the input matrix $\mathbf{A} = \mathbf{a}\mathbf{a}^\intercal$, where $\mathbf{a}$ is the vector as in Theorem 13. Then, a vector-matrix-vector product with $\mathbf{A}$ is $\mathbf{v}^\intercal \mathbf{A} \mathbf{v} = \langle \mathbf{v}, \mathbf{a} \rangle^2$, which is the square of a linear measurement with $\mathbf{a}$. Further, $\text{tr}(\mathbf{A}) = \text{tr}(\mathbf{a}\mathbf{a}^\intercal) = \|\mathbf{a}\|_2^2$. So, we must have that the number of vector-matrix-vector queries made by $\mathbf{A}$ cannot violate the linear measurement lower bound in Theorem 13.

So, our goal is now to prove Theorem 13. The crux of the proof is to use Lemma 8 to show that no Kronecker matrix-vector method can distinguish between linear measurements between two vectors:

**Problem 14.** *Fix* $n, q \in \mathbb{N}$ *and* $\lambda = 6\sqrt{\varepsilon}$. *Let* $\mathbf{g} \in \mathbb{R}^{n^q}$ *be a* $\mathcal{N}(\mathbf{0}, \mathbf{I})$ *vector, and let* $\mathbf{u} = \mathbf{u}_1 \otimes \cdots \otimes \mathbf{u}_q$ *where each* $\mathbf{u}_i \in \mathbb{R}^n$ *is distributed uniformly on the set of vectors with* $\|\mathbf{u}_i\|_2^2 = n$. *Further, let*

$$\mathbf{a}_0 := \mathbf{g} \qquad and \qquad \mathbf{a}_1 := \mathbf{g} + \lambda \mathbf{u}.$$

*Suppose that nature samples* $i \in \{0, 1\}$ *uniformly at random. An algorithm then computes* $t$ *linear measurements with* $\mathbf{a} := \mathbf{a}_i$ *and has to guess if* $\mathbf{a} = \mathbf{a}_0$ *or* $\mathbf{a} = \mathbf{a}_1$.

In Appendix B we formally prove the exponential lower bound against Problem 14 as stated in Theorem 13. We do take a moment to sketch the proof here though.

Consider a non-adaptive Kronecker-structured linear measurement algorithm for Problem 14. If a method is non-adaptive, then we can think of it as a method that picks a matrix $\mathbf{V}$ with Kronecker-structured columns and computes $\mathbf{V}^\intercal \mathbf{a} = [\langle \mathbf{v}^{(1)}, \mathbf{a} \rangle \cdots \langle \mathbf{v}^{(t)}, \mathbf{a} \rangle]$. So, to prove our lower bound against non-adaptive algorithms, we need to show that for all $\mathbf{V}$ with $t$ Kronecker-structured columns and condition number at most $\kappa$, it is not possible to distinguish $\mathbf{w}_0 := \mathbf{V}^\intercal \mathbf{a}_0$ from $\mathbf{w}_1 := \mathbf{V}^\intercal \mathbf{a}_1$.

We start by examining these two distributions. Because $\mathbf{g}$ is Gaussian, we know that $\mathbf{w}_0 \sim \mathcal{N}(\mathbf{0}, \mathbf{V}^\intercal \mathbf{V})$. Similarly, for a fixed value of $\mathbf{u}$, we know that $\mathbf{w}_1 \sim \mathcal{N}(\lambda \mathbf{V}^\intercal \mathbf{u}, \mathbf{V}^\intercal \mathbf{V})$. These two distributions differ only in their means and share the same covariance matrix. In particular, we can easily bound the KL Divergence between $\mathbf{w}_0$ and $\mathbf{w}_1$ for a fixed value of $\mathbf{u}$:

$$D_{KL}(\mathbf{w}_0 \| \mathbf{w}_1 | \mathbf{u}) = \frac{\lambda^2}{2} \mathbf{u}^\intercal \mathbf{V}(\mathbf{V}^\intercal \mathbf{V})^{-1} \mathbf{V}^\intercal \mathbf{u}. \quad (1)$$

This follows from the KL divergence between $\mathcal{N}(\boldsymbol{\mu}, \boldsymbol{\Sigma})$ and $\mathcal{N}(\mathbf{0}, \boldsymbol{\Sigma})$ being exactly $\frac{1}{2}\boldsymbol{\mu}^\intercal \boldsymbol{\Sigma}^{-1}\boldsymbol{\mu}$. We can then use a union bound with Lemma 8 to say that $\|\mathbf{V}^\intercal \mathbf{u}\|_2^2 = \sum_{i=1}^t \langle \mathbf{v}^{(i)}, \mathbf{u} \rangle^2 \leq t C_\tau^{-q}$ with probability at least $1 - t C_0^{-q}$.

So, we can bound

$$D_{KL}(\mathbf{w}_0\|\mathbf{w}_1|\mathbf{u}) = \frac{\lambda^2}{2}\mathbf{u}^\mathsf{T}\mathbf{V}(\mathbf{V}^\mathsf{T}\mathbf{V})^{-1}\mathbf{V}^\mathsf{T}\mathbf{u}$$
$$\leq \frac{\lambda^2}{2}\|\mathbf{V}^\mathsf{T}\mathbf{u}\|_2^2\,\|(\mathbf{V}^\mathsf{T}\mathbf{V})^{-1}\|_2$$
$$\leq \frac{\lambda^2}{2}(tC_\tau^{-q})\kappa^2$$

where the last line uses that we can take the columns of $\mathbf{V}$ to be unit vectors without loss of generality, so that

$$\|(\mathbf{V}^\mathsf{T}\mathbf{V})^{-1}\|_2 = \frac{1}{\sigma_{\min}(\mathbf{V})^2} \leq \frac{\sigma_{\max}(\mathbf{V})^2}{\sigma_{\min}(\mathbf{V})^2} \leq \kappa^2.$$

By Pinsker's Inequality and the Neyman-Pearson Lemma (Csiszár & Körner, 2011; Neyman & Pearson, 1933), we know that $\mathbf{w}_0$ and $\mathbf{w}_1$ cannot be distinguished with probability $\frac{2}{3}$ so long as their KL divergence is at most $O(1)$, which happens when $t = O(\frac{C_\tau^q}{\kappa^2\lambda^2}) = O(\frac{C_\tau^q}{\kappa^2\varepsilon})$. Mixed with the requirement that our union bound earlier hold with high probability, we also require that $t = O(C_0^q)$. This yields our overall lower bound, showing that $\mathbf{w}_0$ and $\mathbf{w}_1$ cannot be distinguished when both $t = O(C_0^q)$ and $t = O(\frac{C_\tau^q}{\kappa^2\varepsilon})$, completing the lower bound.

We note the above lower bound holds only against non-adaptive algorithms. In Appendix B, we adapt a proof strategy from (Simchowitz et al., 2017) to show that adaptivity cannot help much. That proof is much more involved, and the fundamental intuitions unique to our method are well captured by the analysis above. In brief, the proof against adaptive methods shows that at every point of time $i \in [t]$, the algorithm does not suddenly learn new information about the direction of $\mathbf{u}$, owing to Lemma 8. This analysis gives us the benefit of proving a lower bound against adaptive methods, but comes with the downside of having slightly worse rates, giving $t = \Omega(\min\{C_0^{q/2}, \frac{C_\tau^{q/2}}{\kappa^2\sqrt{\varepsilon}}\}$ in the adaptive case as opposed to $t = \Omega(\min\{C_0^q, \frac{C_\tau^q}{\kappa^2\varepsilon}\}$ in the non-adaptive one.

### 4.2. Proof Sketch of Spectral Norm Estimation Lower Bound

In this section, we outline the proof of Theorem 6. We follow the proof strategy in (Simchowitz et al., 2017) again here. In (Simchowitz et al., 2017), the authors show lower bounds against distinguishing between two matrices from matrix-vector products. Specifically, they let $\mathbf{G} \in \mathbb{R}^{D \times D}$ be a matrix with iid $\mathcal{N}(0,1)$ entries and let $\mathbf{u} \in \mathbb{R}^D$ be a random unit vector in $\mathbb{R}^D$. They show that distinguishing between

$$\mathbf{A}_0 = \frac{\mathbf{G} + \mathbf{G}^\mathsf{T}}{\sqrt{2D}} \quad \text{and} \quad \mathbf{A}_1 = \frac{\mathbf{G} + \mathbf{G}^\mathsf{T}}{\sqrt{2D}} + \lambda\mathbf{u}\mathbf{u}^\mathsf{T}$$

requires computing at least $t = \Omega(\frac{\log(D)}{\log(\lambda)})$ classical (non-Kronecker) matrix-vector products. We take $D = n^q$. We abstract out their analysis in Appendix D, allowing us to pick a different distribution over unit vectors $\mathbf{u}$ and restricting the set of matrix-vector query vectors to be Kronecker-structured. Fundamentally, by taking $\mathbf{u}$ to instead be the Kronecker product of iid unit vectors in $\mathbb{R}^n$, we can against take advantage of Lemma 8 much like in trace estimation lower bound of Section 4.1. Intuitively, we again have that the inner products between our query vectors and the planted random vector are exponentially small, and therefore at every time step $i \in [t]$ of the algorithm, it is exceedingly unlikely that the matrix-vector algorithm suddenly goes from having small inner product with $\mathbf{u}$ to having a large inner product with $\mathbf{u}$.

Formally, we prove the following distinguishing lower bound:

**Theorem 15.** *Consider the problem using Kronecker matrix-vector products to test if $\mathbf{A} = \mathbf{A}_0$ or $\mathbf{A} = \mathbf{A}_1$ as shown above, where $\mathbf{u} = \mathbf{u}_1 \otimes \cdots \otimes \mathbf{u}_q$ for iid uniformly random unit vectors $\mathbf{u}_i \in \mathbb{R}^n$. Then, any $\kappa-$conditioned Kronecker matrix-vector algorithm needs at least $t = \Omega(\min\{C_0^{q/2}, \frac{C_\tau^q}{\lambda^2\kappa^2}\})$ queries to correctly identify $\mathbf{A}$ with probability $\frac{2}{3}$.*

We prove Theorem 15 in Appendix C. The key payoff from this testing lower bound comes from comparing the spectral norms of $\mathbf{A}_0$ and $\mathbf{A}_1$. The spectral norm of $\mathbf{A}_0$ is at most $O(1)$ while the spectral norm of $\mathbf{A}_1$ is $\Omega(\lambda)$ for large $\lambda$. In particular, if we take $\lambda = C_\tau^{q/2}$ then we get the following lower bound:

**Corollary 16.** *There exists a number $C > 1$ such that any $\kappa$-conditioned Kronecker matrix-vector algorithm that can determine if $\|\mathbf{A}\|_2 \leq 3$ or $\|\mathbf{A}\|_2 \geq C^q$ with probability at least $\frac{2}{3}$ must use at least $t = \Omega(C_0^{q/2}, \frac{C^q}{\kappa^2})$ queries.*

This means that even computing an overwhelmingly coarse approximation to the spectral norm of a matrix must incur an exponential query complexity. This corollary directly implies Theorem 6.

## 5. Zero Testing

In this section, we consider the zero-testing problem with Kronecker measurements. That is, we suppose that we have a nonzero tensor $\mathcal{A} \in (\mathbb{R}^n)^{\otimes q}$. How many Kronecker structured measurements of the form $v_1 \otimes \ldots \otimes v_q$ do we need to show that $\mathcal{A}$ is nonzero?

As it turns out, the most difficult case for zero-testing is when $\mathcal{A}$ itself has Kronecker structure. When we can write $\mathcal{A} = \mathbf{a}_1 \otimes \ldots \otimes \mathbf{a}_q$, then each measurement of $\mathcal{A}$ gives a result of the form $\prod_i(\mathbf{v}_i^\mathsf{T}\mathbf{a}_i)$, which is 0 as long as at least one of the terms in the product is 0. This suggests

that we should first study the zero-testing problem in the non-Kronecker setting.

Here, we make the additional assumption that the entries of each $\mathbf{a}_i$ come from a fixed "alphabet" that we call $\mathcal{L} \subseteq \mathbb{C}$. This assumption may seem strange at first, but one motivation is that in the non-Kronecker setting, trace estimators such as Hutchinson typically only require that one sketch using Rademacher random vectors. If one attempts to use a Kronecker product of Rademacher vectors, then trace estimation turns out to require a number of measurements that is exponential in $q$. The zero-testing problem gives a simpler setting in which to observe this exponential dependence. Indeed the reason is quite similar to our norm-estimation results – Kronecker products of Rademacher can be orthogonal to a fixed tensor with high probability, just as how Kronecker products of Gaussians are typically very nearly orthogonal to one another.

To set up some notation, suppose we have a tensor $\mathcal{A} \in (\mathbb{R}^n)^{\otimes q}$. For $\mathbf{v} \in \mathbb{R}^n$ say that the measurement of $\mathcal{A}$ along mode $i$ by $\mathbf{v}$ is the tensor in $(\mathbb{R}^n)^{\otimes q-1}$ that results from taking the inner product of $\mathbf{v}$ against each of the mode $i$ fibers. We use the notation $\mathcal{A} \times_i \mathbf{v}$. This is the *Modal Product* as defined in (Golub & Van Loan, 2013).

The following definitions will be useful for writing our upper and lower bounds with respect to given alphabets.

**Definition 17.** *For a given alphabet $\mathcal{L}$, and a field $\mathbb{F}$, either $\mathbb{R}$ or $\mathbb{C}$, let*

$$P_{\mathbb{F}}(\mathcal{L}, n) = \min_{\mathcal{D}} \max_{\mathbf{u} \in \mathcal{L}^n} \Pr_{\mathbf{v} \sim \mathcal{D}} [\mathbf{v}^{\mathsf{T}} \mathbf{u} \neq 0],$$

*where $\mathcal{D}$ ranges over all probability distributions on the nonzero vectors in $\mathbb{F}^n$. When $\mathbb{F}$ is not specified, we assume that $\mathbb{F} = \mathbb{R}$.*

*Similarly, we define*

$$Q_{\mathbb{F}}(\Sigma, n) = \max_{\mathcal{D}_{\mathcal{L}}} \min_{\mathbf{v} \in \mathbb{F}^n} \Pr_{\mathbf{u} \sim \mathcal{D}_{\mathcal{L}}} [\mathbf{v}^{\mathsf{T}} \mathbf{u} \neq 0],$$

*where $\mathcal{D}_{\mathcal{L}}$ ranges over distributions on $\mathcal{L}^n$.*

Intuitively, $P$ captures highest success probability that we can achieve for zero-testing on the hardest input distribution. So upper-bounding $P$ can be used to give a zero-testing lower bound.

Similarly a lower bound on $Q$ shows that there is a distribution over measurements that has good success probability of giving a nonzero measurement on all inputs. So a lower bound on $Q$ can be used to give an upper bound for the zero-testing problem.

**Theorem 18.** *We have the following.*

1. *$P(\{-1, 1\}, 2) \leq \frac{1}{2}$*

2. *For an arbitrary finite alphabet $\mathcal{L}$, $P(\mathcal{L}, n) \leq 1 - \frac{1}{|\mathcal{L}|} \frac{n - |\mathcal{L}|}{n-1}$*

3. *For an arbitrary finite alphabet $\mathcal{L}$, $Q(\mathcal{L}, n) \geq 1 - \frac{1}{|\mathcal{L}|}$*

4. *$P_{\mathbb{C}}(\{-1, 1, i, -i\}, 2) = Q_{\mathbb{C}}(\{-1, 1, i, -i\}, 2) = 3/4$*

*Proof.* 1. Choose $\mathcal{D}$ to be uniform over $\{(1, 1), (1, -1)\}$. Then any vector $\mathbf{u}$ in $\{-1, 1\}^2$ has dot product 0 with one element of $\{(1, 1), (1, -1)\}$. So if $\mathbf{v}$ is uniform from $\{(1, 1), (1, -1)\}$, then with probability $1/2$, $\mathbf{v}^{\mathsf{T}} \mathbf{u} = 0$.

2. Choose $\mathcal{D}$ to be the uniform distribution over vectors of support size 2 whose first nonzero value is 1 and whose second nonzero value is $-1$. Let $\mathbf{v}$ be drawn from $\mathcal{D}$ and let $i, j$ be the coordinates of its support. Now suppose that $\mathbf{u}$ has entries in $\mathcal{L}$. Then $\mathbf{v}^{\mathsf{T}} \mathbf{u} = 0$ precisely when $[\mathbf{u}]_i = [\mathbf{u}]_j$.

For each $k \in \mathcal{L}$, let $n_k$ denote the number of entries of $\mathbf{u}$ that take value $k$. The probability that $[\mathbf{u}]_i = [\mathbf{u}]_j$ is then

$$\binom{n}{2}^{-1} \left( \binom{n_1}{2} + \binom{n_2}{2} + \ldots + \binom{n_L}{2} \right).$$

We can bound this sum as

$$\sum_{i=1}^{|\mathcal{L}|} \binom{n_i}{2} = \frac{1}{2} \sum_{i=1}^{|\mathcal{L}|} (n_i^2 - n_i)$$

$$= \frac{1}{2} \left( \sum_{i=1}^{|\mathcal{L}|} n_i^2 - n \right) \geq \frac{1}{2} \left( \frac{n^2}{|\mathcal{L}|} - n \right).$$

In the last line we used the bound $\sum_{i=1}^{|\mathcal{L}|} n_i^2 \geq \frac{1}{|\mathcal{L}|} \left( \sum_{i=1}^{|\mathcal{L}|} n_i \right)^2$, which is a special case of Cauchy-Schwarz. It follows that

$$\Pr([\mathbf{u}]_i = [\mathbf{u}]_j) \geq \frac{1}{|\mathcal{L}|} \frac{n - |\mathcal{L}|}{n - 1}.$$

3. Choose $\mathcal{D}_{\mathcal{L}}$ to to have i.i.d. entries over $\mathcal{L}$ and let $\mathbf{u}$ be drawn from $\mathcal{D}_{\mathcal{L}}$ Let $i$ be the first nonzero coordinate of $\mathbf{v}$. Conditioned on all coordinates of $\mathbf{u}$ except $i$, the value of $\mathbf{v}^{\mathsf{T}} \mathbf{u}$ is uniform over a set of size $|\mathcal{L}|$. Therfore $\mathbf{v}^{\mathsf{T}} \mathbf{u}$ is 0 with probability at most $\frac{1}{|\mathcal{L}|}$.

4. To bound $P_{\mathcal{C}}$, choose the distribution $\mathcal{D}$ to be uniform over $\{(1, 1), (1, -1)(1, i), (1, -i)\}$. Now observe that any two-dimensional vector with entries in $\{\pm 1, \pm i\}$ is orthogonal to one of these four vectors. So $P_{\mathcal{C}} \leq \frac{3}{4}$.

Similarly, for $Q_{\mathcal{C}}$ we choose our measurement distribution $\mathcal{D}_{\mathcal{L}}$ to be uniform over

$\{(1,1),(1,-1)(1,i),(1,-i)\}$. These vectors are pairwise linearly independent, so any fixed $\mathbf{u}$ is orthogonal to at most one of them. Thus $Q_{\mathcal{C}} \geq 3/4$.

□

The following gives a general lower bound for the zero-testing problem via Kronecker measurements. The idea is effectively to boost the analogous lower bound for non-Kronecker-structured measurements. We also give a corresponding upper bound that works by reducing to the analogous upper bound for non-Kronecker-structured measurements inductively along each mode.

**Theorem 19.** *(i) Zero-testing of an arbitrary vector $\mathbf{v} \in (\mathbb{R}^n)^{\otimes q}$ with $\frac{2}{3}$ success probability, using Kronecker structured measurements in $(\mathcal{L}^n)^{\otimes q}$ requires at least $\frac{2}{3}\Omega(P_{\mathbb{F}}(\mathcal{L}, n)^{-q})$ measurements.*

*(ii) Suppose that $\mathcal{L} \subseteq \mathbb{F}$. There is a zero-tester using Kronecker-structured measurements over the alphabet $\Sigma$, that succeeds with $\frac{2}{3}$ probability and uses $2Q_{\mathbb{F}}(\mathcal{L}, n)^{-q}$ measurements.*

*Proof.* For the lower bound, let $\mathcal{D}$ be the distribution that achieves the minimum in the definition of $p(\mathcal{L}, n)$. Let $\mathbf{v}_1, \ldots, \mathbf{v}_q$ be drawn independently from $\mathcal{D}$. Let $\mathbf{x}_1, \ldots, \mathbf{x}_q$ be arbitrary fixed vectors in $\mathcal{L}^n$. Then we have

$$(\mathbf{x}_1 \otimes \ldots \otimes \mathbf{x}_q)^{\mathsf{T}}(\mathbf{v}_1 \otimes \ldots \otimes \mathbf{v}_q) = (\mathbf{x}_1^{\mathsf{T}}\mathbf{v}_1)\ldots(\mathbf{x}_q^{\mathsf{T}}\mathbf{v}_q).$$

Note that $\mathbf{x}_i^{\mathsf{T}}\mathbf{v}_i \neq 0$ with probability at most $p(\mathcal{L}, n)$. Each of the terms $\mathbf{x}_i^{\mathsf{T}}\mathbf{v}_i$ is independent, and so the probability that the product is nonzero is at most $p(\mathcal{L}, n)^q$.

Suppose that an algorithm makes $m$ Kronecker-structured measurements. Then by a union bound, the probability that at least one of the measurements is nonzero is at most $mp(\mathcal{L}, n)^q$. The claim follows.

For the upper bound, choose our measurement vectors to be of the form $\mathbf{u}_1 \otimes \ldots \otimes \mathbf{u}_q$ where the $\mathbf{u}_i$'s are i.i.d. from the distribution $\mathcal{D}_{\mathcal{L}}$. Then for a nonzero tensor $\boldsymbol{\mathcal{A}}$ have

$$\langle \boldsymbol{\mathcal{A}}, \mathbf{u}_1 \otimes \ldots \otimes \mathbf{u}_q \rangle = \boldsymbol{\mathcal{A}} \times_1 \mathbf{u}_1 \times_2 \mathbf{u}_2 \ldots \times_q \mathbf{u}_q.$$

Since $\boldsymbol{\mathcal{A}}$ is nonzero, $\boldsymbol{\mathcal{A}}$ has some nonzero fiber along mode 1, and therefore $\boldsymbol{\mathcal{A}} \times_1 \mathbf{u}_1$ is nonzero with probability at least $Q_{\mathbb{F}}(\mathcal{L}, n)$. Continuing inductively, the measurement above is nonzero with probability at least $Q_{\mathbb{F}}(\mathcal{L}, n)^q$. Given $m$ measurements of this form, the probability that all of them are 0 is at most

$$(1 - Q_{\mathbb{F}}(\mathcal{L}, n)^q)^m \leq \exp(-mQ_{\mathbb{F}}(\mathcal{L}, n)^q),$$

which is at most $1/4$ for $m \geq 2Q_{\mathbb{F}}(\mathcal{L}, n)^{-q}$. □

**Lemma 20.** *An algorithm that performs constant-factor trace estimation requires at least $\frac{2}{3}P_{\mathbb{F}}(\mathcal{L}, n)^{-q}$ Kronecker-structured vector-matrix-vector queries.*

*Proof.* Let $\mathbf{x}_1, \ldots, \mathbf{x}_q$ be drawn from the distribution $\mathcal{D}$ in the definition of $P_{\mathbb{F}}(\mathcal{L}, n)$. Set $\mathbf{x} = \mathbf{x}_1 \otimes \ldots \otimes \mathbf{x}_q$. Take our matrix $\mathbf{A}$ to be $\mathbf{x}\mathbf{x}^{\mathsf{T}}$.

Suppose that we make $t$ measurements of the form $\mathbf{A}\mathbf{v}^{(i)}$, where $\mathbf{v}^{(i)}$ for $i \in [t]$ has Kronecker structure and uses the alphabet $\mathcal{L}$. The result of the measurement is nonzero precisely when $\mathbf{x}^{\mathsf{T}}\mathbf{v}^{(i)} \neq 0$. The probability that this is nonzero is exactly $(P_{\mathbb{F}}(\mathcal{L}, n))^q$. By a union bound, the probability that at least one of the matrix-vector products is nonzero is at most $t(P_{\mathbb{F}}(\mathcal{L}, n))^q$. On the other hand, a constant factor trace estimator must distinguish $\mathbf{A}$ from the $\mathbf{0}$ matrix with probability $\frac{2}{3}$, so we need $t(P_{\mathbb{F}}(\mathcal{L}, n))^q \geq \frac{2}{3}$. from which the claim follows. □

Combining the previous results give the following bounds for zero-testing.

**Corollary 21.** *Zero testing for a tensor in $(\mathbb{R}^n)^{\otimes q}$ with success probability $\frac{2}{3}$*

1. *requires at least $\Omega(2^q)$ measurements over the alphabet $\{-1, 1\}$ when $n = 2$.*

2. *Requires at least $\Omega((1 - \frac{1}{|\mathcal{L}|}\frac{n-|\mathcal{L}|}{n-1})^q)$ for an arbitrary alphabet $\mathcal{L}$.*

*For zero testing for a tensor in $(\mathbb{C}^n)^{\otimes q}$, it is necessary and sufficient to use $\Theta((4/3)^q)$ measurements for the alphabet $\{-1, 1, i, -i\}$ when $n = 2$.*

*Proof.* Combine Theorem 18 with Theorem 19. □

We obtain a similar corollary for trace estimation.

**Corollary 22.** *Constant-factor trace estimation of a real PSD matrix requires $\Omega(2^q)$ measurements when $n = 2$ and when using Rademacher Kronecker-structured matrix-vector queries, i.e. with vectors in $(\{-1, 1\}^2)^{\otimes q}$.*

*Constant-factor trace estimation of a complex PSD matrix requires $\Omega((4/3)^q)$ measurements when $n = 2$ and when using complex Rademacher Kronecker-structured matrix-vector queries, i.e. with vectors in $(\{-1, 1, i, -i\}^2)^{\otimes q}$.*

*Proof.* Combine Corollary 21 with Lemma 20. □

## 6. Conclusion

We addressed several fundamental linear algebraic problems in the Kronecker matrix-vector query model. A number of interesting questions remain. Some of Our lower bounds have a dependence on the condition number of the measurement matrix. Is this dependence necessary? This is open even in the case of non-adaptive measurements. Rigorously, we know that proving the following corollary would

suffice to remove the conditioning assumption in both the non-adaptive and adaptive cases:

**Conjecture 23.** *Let* $\mathbf{u} = \mathbf{u}_1 \otimes \cdots \otimes \mathbf{u}_q$ *where each* $\mathbf{u}_i$ *is a uniformly random unit vector in* $\mathbb{R}^n$. *Let* $\mathbf{V} = [\mathbf{v}^{(1)} \cdots \mathbf{v}^{(t)}]$ *where each* $\mathbf{v}^{(i)}$ *is an arbitrary (non-random) Kronecker-structured vector. Let* $\mathbf{P}$ *be the orthogonal projection onto the range of* $\mathbf{V}$. *Then, so long as* $t \leq \text{poly}(n, q)$, *we have that*

$$\|\mathbf{Pu}\|_2^2 \leq \frac{c_1^{-q}}{n^q} \qquad \text{with probability at least } 1 - c_2^{-q}$$

*for some* $c_1, c_2 > 1$.

The above conjecture is a direct generalization of Lemma 8. To see this, note that taking $t = 1$ in Conjecture 23 exactly recovers Lemma 8.

In the case that Conjecture 23 does not hold, this suggests that a ill-conditioned might be efficient in the Kronecker matrix-vector model. Namely, does there exist a Khatri-Rao sketching matrix that allows for $\ell_2$ norm estimation (and is extremely poorly conditioned)? It would also be interesting to obtain tight bounds for trace estimation in the Kronecker matrix-vector model. Lower bounds for Hutchinson-style estimators are known, but could there be better estimators, perhaps analogous to the Hutch++ (Meyer et al., 2021) algorithm?

## Acknowledgments

Raphael Meyer was partially supported by a Caltech Center for Sensing to Intelligence grant to Joel A. Tropp and ONR Award N-00014-24-1-2223 to Joel A. Tropp. William Swartworth and David Woodruff received support from a Simons Investigator Award and Office of Naval Research award number N000142112647. We thank Ethan Epperly for detailed comments improving the presentation of the paper.

## Impact Statement

This paper presents work whose goal is to advance the field of Machine Learning. There are many potential societal consequences of our work, none which we feel must be specifically highlighted here.

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

## A. Near-Total Orthogonality with Real Vectors

In this section, we prove Lemma 8 and related concentrations and lemmas that characterize the near-total orthogonality of the Kronecker product of random unit vectors with respect to otehr Kronecker-structured vectors. We conclude with a short lemma showing how conditioning relates to projections of Kronecker-structured vectors.

**Lemma 24.** *Let $X$ be disitributed as the first entry of a uniformly random vector in $\sqrt{n}\mathbb{S}^n$. Let $Y = \log|X|$. Then $Y$ is subexponential with subexponential norm $\|Y\|_{\psi_1} \le O(1)$.*

*Proof.* Recall that the $\beta(\frac{1}{2}, \frac{n-1}{2})$ distribution has pdf given by

$$\frac{\Gamma(n/2)}{\Gamma(1/2)\Gamma((n-1)/2)} x^{-1/2}(1-x)^{(n-3)/2} := f(x)$$

on the interval $[0, 1]$, and that $\frac{1}{\sqrt{n}}|X|$ is distributed as the square root of a $\beta(1/2, \frac{n-1}{2})$ random variable.

By the change of variables formula, the pdf of $\frac{1}{\sqrt{n}}|X|$ is given by

$$f(x^2) \cdot \frac{d}{dx} x^2 = 2 \frac{\Gamma(n/2)}{\Gamma(1/2)\Gamma((n-1)/2)}(1-x)^{(n-3)/2},$$

which is is uniformly bounded by $C\sqrt{n}$ on $[0, 1/2]$ for an absolute constant $C$.

We then have that for $t \ge \log 2$ that,

$$\Pr(Y \le -t) = \Pr(|X| \le e^{-t}) = \Pr(\frac{1}{\sqrt{n}}|X| \le \frac{1}{\sqrt{n}}e^{-t}) \le \frac{2}{\sqrt{n}}e^{-t} \sup_{x \in [0,1/2]} f_X(x) \le 2Ce^{-t}.$$

Also $X$ is subgaussian with constant subgaussian norm indedendent of $n$ (see for example Theorem 3.4.6 in (Vershynin, 2018).) Thus $X$ is also subexponential with constant subexponential norm. So for positive $t$, $X$ satisfies a right tail bound of the form

$$\Pr(X \ge t) \le \exp(-ct).$$

Since $Y \le X$, we obtain the same right tail bound for $Y$, and our claim follows. $\qquad\square$

**Lemma 8** **Restated.** *Let $\mathbf{u} = \mathbf{u}_1 \otimes \cdots \otimes \mathbf{u}_q$ where $\mathbf{u}_i$ is a uniformly random unit vector in $\mathbb{R}^n$. Then, for any kronecker-strucutred unit vector $\mathbf{v} = \mathbf{v}_1 \otimes \cdots \otimes \mathbf{v}_q$ we have that $\tau \le C_\tau^{-q}$ has*

$$f(\tau) := \Pr\left[\langle \mathbf{u}, \mathbf{v}\rangle^2 \ge \frac{\tau}{n^q}\right] \le C_0^{-q}$$

*for some universal constants $C_\tau, C_0 > 1$.*

*Proof.* We start by letting $X := \langle \mathbf{u}, \mathbf{v}\rangle^2$, $X_i := \langle \mathbf{u}_i, \mathbf{v}_i\rangle^2$, and $Y_i := \ln(X_i)$, so that $Y := \ln(X) = \sum_{i=1}^{q} Y_i$ is a sum of iid terms. We will argue the concentration of $X$ via the concentration of $Y$. By Lemma 24, we know that $\log|Z|$ has sub-exponential norm $K$, where $Z$ is the first entry of a random on the unit sphere of radius $\sqrt{n}$. Since the mean of $\log|Z|$ is at most $1.32 + \frac{1}{n} \le 1.4$ for $n \ge 13$, we know that $\log|Z| - \mathbb{E}[\log|Z|]$ has sub-exponential norm at most $K + 1.4$. Then, by Bernstein's Inequality (as written in Proposition 4.2 of (Zhang & Chen, 2020)),

$$\Pr\left[\sum_{i=1}^{q} \log|Z_i| \ge q\,\mathbb{E}[\log|Z_i|] + 2t\right] \le e^{-\frac{1}{4}\min\{\frac{t^2}{8q(K+1.4)^2}, \frac{t}{2(K+1.4)}\}}$$

Since $Y_i = 2\log|Z_i| - \log(n)$, we can equivalently take $\mu := \mathbb{E}[Y_i]$ and write

$$\Pr[\sum_{i=1}^{q} Y_i \ge q\mu + t] \le e^{-\frac{1}{4}\min\{\frac{t^2}{8q(K+1.4)^2}, \frac{t}{2(K+1.4)}\}}$$

Recalling that $X = e^{\sum_i Y_i}$ and that $\mu \leq 0$,

$$\Pr\left[X \geq e^{t-q|\mu|}\right] \leq e^{-\frac{1}{4}\min\{\frac{t^2}{8q(K+1.4)^2}, \frac{t}{2(K+1.4)}\}}$$

Next we need to compute $\mu = \mathbb{E}[Y_i] = \mathbb{E}[\log(X_i)]$. Letting $\psi$ denote the digamma function, we can write $\mathbb{E}[\log(X_i)] = \psi(\alpha) - \psi(\alpha + \beta) = \psi(\frac{1}{2}) - \psi(\frac{n}{2})$, and therefore that

$$1.27 + \ln(n) - \frac{2}{n} \leq |\mu| \leq 1.271 + \ln(n)$$

Then, we know that $X \geq e^{t-q|\mu|}$ implies that $X \geq e^{t-q(1.271+\ln(n))} = n^{-q}e^{t-1.271q}$. So, we have

$$\Pr\left[X \geq \frac{e^{t-1.271q}}{n^q}\right] \leq e^{-\frac{1}{4}\min\{\frac{t^2}{8q(K+1.4)^2}, \frac{t}{2(K+1.4)}\}}$$

Taking $t = 16(K+1.4)^2(\sqrt{1 + \frac{1.271}{8(K+1.4)^2}} - 1)q$ then gives us

$$\Pr\left[X \geq \frac{e^{-\alpha q}}{n^q}\right] \leq e^{-\alpha q}$$

where $\alpha = 1.271 - 16(K+1.4)^2(\sqrt{1 + \frac{1.271}{8(K+1.4)^2}} - 1) \in (0, 0.006)$. From Lemma 24, we know that $K = O(1)$, which completes the proof.

$\square$

We will also need the following result on the MGF of the inner product of Kronecker-structured vectors.

**Lemma 25.** *Let* $\mathbf{u} = \mathbf{u}_1 \otimes \cdots \otimes \mathbf{u}_q$ *where* $\mathbf{u}_i$ *is a uniformly random unit vector in* $\mathbb{R}^n$. *Then, for any kronecker-strucutred unit vector* $\mathbf{v} = \mathbf{v}_1 \otimes \cdots \otimes \mathbf{v}_q$ *and* $\eta \in (0, 1)$,

$$\mathbb{E}[e^{\eta|\langle \mathbf{u}, \boldsymbol{\theta}\rangle|}] \leq 1 + \frac{2\eta}{n^q} \leq e^{2\eta n^{-q}}.$$

*Proof.* We approach this bound via linearization. Since $\eta|\langle \mathbf{u}, \boldsymbol{\theta}\rangle| \leq \eta \leq 1$, we know that $e^{\eta|\langle \mathbf{u}, \boldsymbol{\theta}\rangle|} \leq 1 + 2\eta|\langle \mathbf{u}, \boldsymbol{\theta}\rangle|$. So, we bound

$$\mathbb{E}[e^{\eta|\langle \mathbf{u}, \boldsymbol{\theta}\rangle|}] \leq 1 + 2\eta\,\mathbb{E}[|\langle \mathbf{u}, \boldsymbol{\theta}\rangle|]$$
$$= 1 + 2\eta(\mathbb{E}[|\langle \mathbf{u}_1, \boldsymbol{\theta}_1\rangle|])^q$$

Since $\langle \mathbf{u}_1, \boldsymbol{\theta}_1\rangle$ is a distributed as a $Beta(\frac{1}{2}, \frac{n-1}{2})$ random variable, and since $\langle \mathbf{u}_1, \boldsymbol{\theta}_1\rangle \geq 0$, we know that $\mathbb{E}[|\langle \mathbf{u}_1, \boldsymbol{\theta}_1\rangle|] = \mathbb{E}[\langle \mathbf{u}_1, \boldsymbol{\theta}_1\rangle] = \frac{1}{n}$. So,

$$\mathbb{E}[e^{\eta|\langle \mathbf{u}, \boldsymbol{\theta}\rangle|}] \leq 1 + \frac{2\eta}{n^q} \leq e^{2\eta n^{-q}}$$

where the last inequality uses that $1 + x \leq e^x$ for $x \leq 1$. $\square$

Lastly, we show the following lemma that relates conditioning to the constants $C_0$ and $C_\tau$ from Lemma 8.

**Lemma 26.** *Let* $\mathbf{v}^{(1)}, \cdots, \mathbf{v}^{(t)} \in \mathbb{R}^{n^q}$ *be unit vectors. Suppose that* $\mathbf{V} = [\mathbf{v}^{(1)} \cdots \mathbf{v}^{(t)}] \in \mathbb{R}^{n^q \times t}$ *has condition number less than* $\kappa$. *Let* $\mathbf{X} = [\mathbf{x}^{(1)} \cdots \mathbf{x}^{(t)}] \in \mathbb{R}^{n^q \times t}$ *be an orthogonal matrix that spans* $\mathbf{V}$. *Then, for any unit vector* $\mathbf{u}$, *we have*

$$|\langle \mathbf{x}^{(i)}, \mathbf{u}\rangle|^2 \leq \kappa^2 \|\mathbf{V}^\mathsf{T}\mathbf{u}\|_2^2.$$

*Proof.* There exists some invertible map $\mathbf{R}$ such that $\mathbf{V} = \mathbf{X}\mathbf{R}$ (for instance, if we built $\mathbf{X}$ as the Q factor of the QR of $\mathbf{V}$). Letting $\mathbf{V} = \mathbf{U}\boldsymbol{\Sigma}\mathbf{Z}^\mathsf{T}$ be the SVD of $\mathbf{V}$, notice that

$$\mathbf{R} = \mathbf{X}^\mathsf{T}\mathbf{V} = (\mathbf{X}^\mathsf{T}\mathbf{U})\boldsymbol{\Sigma}\mathbf{Z}^\mathsf{T}$$

is also an SVD and therefore that $\mathbf{R}$ has the same singular values as $\mathbf{V}$. Since $\mathbf{x}^{(i)} = \mathbf{X}\mathbf{e}_i = \mathbf{V}\mathbf{R}^{-1}\mathbf{e}_i$ where $\mathbf{e}_i$ is the $i^{th}$ standard basis vector, we can bound

$$\langle \mathbf{u}, \mathbf{x}^{(i)} \rangle^2 = (\mathbf{u}^\intercal \mathbf{V}\mathbf{R}^{-1}\mathbf{e}_i)^2 \leq \|\mathbf{u}^\intercal \mathbf{V}\|_2^2 \|\mathbf{R}^{-1}\|_2^2 \|\mathbf{e}_i\|_2^2 = \frac{1}{(\sigma_{\min}(\mathbf{V}))^2} \|\mathbf{V}^\intercal \mathbf{u}\|_2^2$$

where we use that $\mathbf{R}$ and $\mathbf{V}$ share singular values in the last equality. Next, since $\mathbf{V}$ has unit vector columns, we know that $\sigma_{\max}(\mathbf{V}) \geq 1$. So, $\frac{1}{(\sigma_{\min}(\mathbf{V}))^2} \leq \frac{(\sigma_{\max}(\mathbf{V}))^2}{(\sigma_{\min}(\mathbf{V}))^2} = \kappa^2$. Therefore, we have

$$\langle \mathbf{u}, \mathbf{x}^{(i)} \rangle^2 \leq \kappa^2 \|\mathbf{V}^\intercal \mathbf{u}\|_2^2$$

completing the lemma. $\qquad\square$

## B. L2 Estimation via Linear Measurements

**Problem 27.** *Fix a vector* $\mathbf{a} \in \mathbb{R}^{n^q}$. *Then, the* Kronecker-structured linear measurement oracle *for* $\mathbf{a}$ *is the oracle that, when given any Kronecker structured vector* $\mathbf{v} \in \mathbb{R}^{n^q}$, *returns* $\langle \mathbf{a}, \mathbf{v} \rangle$. *In the* L2 Estimation via Kronecker Measurements *problem, we have to use a few oracle queries as possible to return a number* $z \in \mathbb{R}$ *such that* $(1-\varepsilon)\|\mathbf{a}\|_2^2 \leq z \leq (1+\varepsilon)\|\mathbf{a}\|_2^2$ *with probability* $\frac{2}{3}$.

**Theorem 28.** *Any (possibly adaptive) algorithm* $\mathcal{A}$ *that solves* Problem 27 *with probability* $\frac{2}{3}$ *using* $\kappa-$*conditioned Kronecker-structured queries must use at least* $t = O(\min\{C_0^{q/2}, \frac{C_\tau^{q/2}}{\kappa^2\sqrt{\varepsilon}}\})$ *queries.*

Our proof methodology mirrors that of Section 6 in (Simchowitz et al., 2017), but applied to this linear measurements framework instead of the matrix-vector framework as studied in their paper (and partially explained in Appendix D). The crux of this section is to show that Lemma 8 implies the lower bound in Theorem 28. We prove this lower bound by appealing to the following testing problem:

**Problem 29.** *Fix* $n, q \in \mathbb{N}$ *and* $\lambda > 1$. *Let* $\mathbf{g} \in \mathbb{R}^{n^q}$ *be a* $\mathcal{N}(\mathbf{0}, \mathbf{I})$ *vector, and let* $\mathbf{u} = \mathbf{u}_1 \otimes \cdots \otimes \mathbf{u}_q$ *where each* $\mathbf{u}_i \in \mathbb{R}^n$ *vector is uniformly distributed on the set of vectors with* $\|\mathbf{u}_i\|_2^2 = n$. *Further, let*

$$\mathbf{a}_0 = \mathbf{g} \qquad and \qquad \mathbf{a}_1 = \mathbf{g} + \lambda\mathbf{u}.$$

*Suppose that nature samples* $i \in \{0, 1\}$ *uniformly at random. Then, an algorithm* $\mathcal{A}$ *computes* $t$ *linear measurements with* $\mathbf{a} := \mathbf{a}_i$ *and then guesses if* $i = 0$ *or* $i = 1$.

The result Theorem 28 follows from combining two results: showing that any L2 estimating algorithm can distinguish $\mathbf{a}_0$ from $\mathbf{a}_1$, and that distinguishing $\mathbf{a}_0$ from $\mathbf{a}_1$ requires exponential query complexity. We start with the former result.

**Lemma 30.** *Let* $\mathcal{A}$ *be any linear measurement algorithm that can solve* Problem 27 *with probability* $\frac{2}{3}$ *for some* $\varepsilon \in (0, 0.25)$. *Then* $\mathcal{A}$ *can solve* Problem 29 *when* $\lambda = 6\sqrt{\varepsilon}$ *and* $n^q = \Omega(\frac{1}{\varepsilon^2})$ *with probability at least* $\frac{3}{5}$.

*Proof.* Throughout this proof, we let $C > 0$ be a large enough constant that both of the concentrations required simultaneously hold with probability $\frac{9}{10}$. We will concretely assume that $\frac{1}{n^{q/2}} \leq \min\{\frac{\lambda^2}{4C}, \frac{\lambda}{8C}\}$. Note that $\|\mathbf{g}\|_2^2$ is a chi-squared random variable with parameter $n^q$. So, $\|\mathbf{a}_0\|_2^2 = \|\mathbf{g}\|_2^2 \in (1 \pm \frac{C}{n^{q/2}})n^q \subseteq (1 \pm \frac{\lambda^2}{4})n^q$. We also know that $\|\mathbf{h}\|_2^2 = n^q$ exactly. Further, since $\mathbf{g}$ is Gaussian, we know that $\langle \mathbf{g}, \mathbf{h} \rangle \sim \mathcal{N}(0, \|\mathbf{h}\|_2^2) = \mathcal{N}(0, n^q)$, and therefore that $|\langle \mathbf{g}, \mathbf{h} \rangle| \leq Cn^{q/2}$. This lets us expand

$$\begin{aligned}
\|\mathbf{a}_1\|_2^2 &= \|\mathbf{g}\|_2^2 + \lambda^2\|\mathbf{h}\|_2^2 - 2\lambda\langle \mathbf{g}, \mathbf{h} \rangle \\
&\geq (1 - \tfrac{\lambda^2}{4})n^q + \lambda^2 n^q - \tfrac{2\lambda C}{n^{q/2}}n^q \\
&\geq (1 - \tfrac{\lambda^2}{4})n^q + \lambda^2 n^q - \tfrac{\lambda^2}{4}n^q \\
&= (1 + \tfrac{\lambda^2}{2})n^q
\end{aligned}$$

So, we have that

$$\|\mathbf{a}_0\|_2^2 \leq (1 + \tfrac{\lambda^2}{4})n^q \qquad \text{and} \qquad \|\mathbf{a}_1\|_2^2 \geq (1 + \tfrac{\lambda^2}{2})n^q.$$

In particular, since $\lambda = 6\sqrt{\varepsilon}$, we have that

$$(1 + \varepsilon) \|\mathbf{a}_0\|_2^2 \le (1 + \varepsilon)(1 + \tfrac{36\varepsilon}{4})n^q < (1 - \varepsilon)(1 + \tfrac{36\varepsilon}{2})n^q \le (1 - \varepsilon) \|\mathbf{a}_1\|_2^2$$

holds for all $\varepsilon \in (0, 0.25)$. In particular, this means that any algorithm $\mathcal{A}$ that can estimate $(1 \pm \varepsilon) \|\mathbf{a}\|_2^2$ from $t$ measurements can distinguish $\mathbf{a}_0$ from $\mathbf{a}_1$ with high probability, completing the proof $\qquad\square$

Next, we show the crux of the lower bound – that Problem 29 has exponential sample complexity lower bound. We show this by applying Imported Lemma 43 to our setting. In order to instantiate this theorem though, we have to introduce some further notation.

**Setting 31.** *Fix an algorithm $\mathcal{A}$ that solves Problem 29. Let $\mathbf{v}^{(1)}, \ldots, \mathbf{v}^{(t)}$ be the (possibly adaptive) query vectors computed by $\mathcal{A}$. Let $w_1, \ldots, w_t$ be the responses from the oracle. That is, $w_i = \langle \mathbf{v}^{(i)}, \mathbf{a} \rangle$. Let $\mathcal{Z}_i = (\mathbf{v}^{(1)}, w_1, \ldots, \mathbf{v}^{(i)}, w_i)$ be the transcript of all information sent between $\mathcal{A}$ and the oracle in the first $i$ queries. By Yao's minimax principle, we assume without loss of generality that $\mathcal{A}$ is deterministic. Therefore, $\mathbf{v}^{(i)}$ is a deterministic function of $\mathcal{Z}_{i-1}$.*

*We let $\mathbb{Q}$ denote the distribution of $\mathcal{Z}_t$ when $\mathbf{a} = \mathbf{a}_0$. We let $\mathbb{P}_{\mathbf{u}}$ denote the distribution of $\mathcal{Z}_t$ when $\mathbf{a} = \mathbf{a}_1$ conditioned on a specific value of $\mathbf{u}$. We let $\bar{\mathbb{P}}$ denote the marginal distribution of $\mathbb{P}_{\mathbf{u}}$ over all $\mathbf{u}$, or equivalently that $\bar{\mathbb{P}}$ is the distribution of $\mathcal{Z}_t$ when $\mathbf{a} = \mathbf{a}_1$. Lastly, we let $A_{\mathbf{u}}^i$ be the event that $\{\forall j \in [i], \langle \mathbf{v}^j, \mathbf{u} \rangle^2 \le \tau_j\}$ for some numbers $0 \le \tau_1 \le \ldots \le \tau_t$ that will be clear from context.*

Following this notation, to show that no algorithm can distinguish $\mathbf{a}_0$ from $\mathbf{a}_1$, it suffices to show that there is low total variation between $\bar{\mathbb{P}}$ and $\mathbb{Q}$. We will do this by applying Imported Lemma 43. In particular, specialized to our context, the lemma says the following:

**Corollary 32.** *Consider Setting 31. Fix any numbers $0 \le \tau_1 \le \ldots \le \tau_t$. If we are given that*

$$\Pr[\exists i \in [t] \; : \; \langle \mathbf{v}^{(i)}, \mathbf{u} \rangle^2 \ge \tau_i] \le z \tag{2}$$

*and that*

$$\mathop{\mathbb{E}}_{\mathcal{Z}_t \sim \mathbb{Q}} \left[ \left( \frac{\mathbb{E}_{\mathbf{u}}[d\mathbb{P}_{\mathbf{u}}(\mathcal{Z}_t \cap A_{\mathbf{u}}^t)]}{d\mathbb{Q}(\mathcal{Z}_t)} \right)^2 \right] \le 1 + z \tag{3}$$

*then the total variation distance between $\bar{\mathbb{P}}$ and $\mathbb{Q}$ is at most $\sqrt{3z}$. In particular, if we take $z = \frac{1}{27}$ then $\mathcal{A}$ cannot distinguish between $\mathbf{a}_0$ and $\mathbf{a}_1$ with probability at least $\frac{2}{3}$.*

This corollary follows directly from plugging in Setting 31 into Imported Lemma 43. In order to prove Theorem 28, we just need to prove that both Equations (2) and (3) hold with $z = \frac{1}{27}$. This will be the focus of the rest of the subsection.

First, we will need the following claim about divergences:

**Lemma 33.** *Let $\mathbb{P}_a$ denote the distribution $\mathcal{N}(\mathbf{a}, \boldsymbol{\Sigma})$, $\mathbb{P}_b$ the distribution $\mathcal{N}(\mathbf{b}, \boldsymbol{\Sigma})$, and $\mathbb{Q}$ the distribution $\mathcal{N}(\mathbf{0}, \boldsymbol{\Sigma})$. Then,*

$$\mathop{\mathbb{E}}_{\mathbf{z} \sim \mathbb{Q}} \left[ \left( \frac{d\mathbb{P}_a(\mathbf{z})}{d\mathbb{Q}(\mathbf{z})} \right)^2 \right] = e^{\mathbf{a}^\intercal \boldsymbol{\Sigma}^{-1} \mathbf{a}}$$

*and*

$$\mathop{\mathbb{E}}_{\mathbf{z} \sim \mathbb{Q}} \left[ \frac{d\mathbb{P}_a(\mathbf{z}) d\mathbb{P}_b(\mathbf{z})}{(d\mathbb{Q}(\mathbf{z}))^2} \right] = e^{\mathbf{a}^\intercal \boldsymbol{\Sigma}^{-1} \mathbf{b}}$$

*Proof.* We prove only the second claim as the first claim follows from taking $\mathbf{b} = \mathbf{a}$. We directly expand the expectation using the corresponding PDFs, nothing that the terms outside the expectation all exactly cancel since our distributions all

share the same covariance matrix.

$$\mathop{\mathbb{E}}_{\mathbf{z}\sim\mathbb{Q}}\left[\frac{d\mathbb{P}_a(\mathbf{z})d\mathbb{P}_b(\mathbf{z})}{(d\mathbb{Q}(\mathbf{z}))^2}\right] = \mathop{\mathbb{E}}_{\mathbf{z}\sim\mathbb{Q}}\left[e^{-\frac{1}{2}(\mathbf{z}-\mathbf{a})^\mathsf{T}\boldsymbol{\Sigma}^{-1}(\mathbf{z}-\mathbf{a})-\frac{1}{2}(\mathbf{z}-\mathbf{b})^\mathsf{T}\boldsymbol{\Sigma}^{-1}(\mathbf{z}-\mathbf{b})+\mathbf{z}^\mathsf{T}\boldsymbol{\Sigma}^{-1}\mathbf{z}}\right]$$

$$= \mathop{\mathbb{E}}_{\mathbf{z}\sim\mathbb{Q}}\left[e^{-\frac{1}{2}(\mathbf{a}^\mathsf{T}\boldsymbol{\Sigma}^{-1}\mathbf{a}+\mathbf{b}^\mathsf{T}\boldsymbol{\Sigma}^{-1}\mathbf{b})+(\mathbf{z}^\mathsf{T}\boldsymbol{\Sigma}^{-1}\mathbf{a}+\mathbf{z}^\mathsf{T}\boldsymbol{\Sigma}^{-1}\mathbf{b})}\right]$$

$$= e^{-\frac{1}{2}(\mathbf{a}^\mathsf{T}\boldsymbol{\Sigma}^{-1}\mathbf{a}+\mathbf{b}^\mathsf{T}\boldsymbol{\Sigma}^{-1}\mathbf{b})}\mathop{\mathbb{E}}_{\mathbf{z}\sim\mathbb{Q}}\left[e^{\mathbf{z}^\mathsf{T}(\boldsymbol{\Sigma}^{-1}(\mathbf{a}+\mathbf{b}))}\right]$$

$$= e^{-\frac{1}{2}(\mathbf{a}^\mathsf{T}\boldsymbol{\Sigma}^{-1}\mathbf{a}+\mathbf{b}^\mathsf{T}\boldsymbol{\Sigma}^{-1}\mathbf{b})}e^{\frac{1}{2}(\boldsymbol{\Sigma}^{-1}(\mathbf{a}+\mathbf{b}))^\mathsf{T}\boldsymbol{\Sigma}(\boldsymbol{\Sigma}^{-1}(\mathbf{a}+\mathbf{b}))} \qquad \text{(Gaussian MGF)}$$

$$= e^{-\frac{1}{2}(\mathbf{a}^\mathsf{T}\boldsymbol{\Sigma}^{-1}\mathbf{a}+\mathbf{b}^\mathsf{T}\boldsymbol{\Sigma}^{-1}\mathbf{b})}e^{\frac{1}{2}(\mathbf{a}+\mathbf{b})^\mathsf{T}\boldsymbol{\Sigma}^{-1}(\mathbf{a}+\mathbf{b})}$$

$$= e^{-\frac{1}{2}(\mathbf{a}^\mathsf{T}\boldsymbol{\Sigma}^{-1}\mathbf{a}+\mathbf{b}^\mathsf{T}\boldsymbol{\Sigma}^{-1}\mathbf{b})}e^{\frac{1}{2}(\mathbf{a}^\mathsf{T}\boldsymbol{\Sigma}^{-1}\mathbf{a}+\mathbf{b}^\mathsf{T}\boldsymbol{\Sigma}^{-1}\mathbf{b}+2\mathbf{a}^\mathsf{T}\boldsymbol{\Sigma}^{-1}\mathbf{b})}$$

$$= e^{\mathbf{a}^\mathsf{T}\boldsymbol{\Sigma}^{-1}\mathbf{b}}$$

$\square$

We will also need the following information-theoretic claim from (Simchowitz et al., 2017):

**Imported Theorem 34** (Proposition 5.1 of (Simchowitz et al., 2017))**.** *Let $\mathcal{P}$ be a prior distribution over parameters $\theta \in \Theta$. Let $\{\mathbb{P}_\theta\}_{\theta\in\Theta}$ be a family of distributions on space $(\mathcal{X}, \mathcal{F})$ parameterized by $\theta$. Let $\{A_\theta\}_{\theta\in\Theta}$ be a set of events defined on $\mathcal{F}$. Let $\mathcal{V}$ be an action space (i.e. an arbitrary set). Let $\mathcal{L} : \mathcal{V} \times \Theta \to \{0,1\}$ be a binary loss function. Let $\mathcal{A}$ denote a determinstic algorithm that observes data and picks an action; that is $\mathcal{A}$ is any map from $\mathcal{X}$ to $\mathcal{V}$. Let $V_0$ be the probability an algorithm can achieve loss 0 without observing data:*

$$V_0 = \sup_{v\in\mathcal{V}} \Pr_{\theta\sim\mathcal{P}}[\mathcal{L}(v, \theta) = 0],$$

*and let $V_v$ be the probability that $\mathcal{A}$ achieves loss 0 after observing a sample from $\mathbb{P}_\theta$ while event $A_\theta$ happens:*

$$V_v = \mathop{\mathbb{E}}_{\theta\sim\mathcal{P}} \Pr_{x\sim\mathbb{P}_\theta}[\mathcal{L}(\mathcal{A}(x), \theta) = 0, A_\theta].$$

*Then, for any probability distribution $\mathbb{Q}$ also on $(\mathcal{X}, \mathcal{F})$,*

$$V_v \le V_0 + \sqrt{V_0(1-V_0)\mathop{\mathbb{E}}_{\theta\sim\mathcal{P}}\mathop{\mathbb{E}}_{x\sim\mathbb{Q}}\left[\left(\frac{d\mathbb{P}_\theta[x]}{d\mathbb{Q}[x]}\right)^2\mathbb{1}_{[A_\theta]}\right]}.$$

This result will suffice to bound Equation (2):

**Lemma 35.** *Consider Setting 31, where $\tau_1 = \cdots = \tau_t = C_\tau^{-q}$. Then, we have that*

$$\Pr[\exists i \in [t] \ : \ \langle\mathbf{v}^{(i)}, \mathbf{u}\rangle^2 \ge \tau_i] \le \tfrac{1}{27}$$

*so long as $t = O(\min\{C_0^{q/2}, \frac{C_\tau^{q/2}}{\kappa^2\sqrt{\varepsilon}}\})$.*

*Proof.* We start by expanding the target probability into a probabilistic claim for each query made by the algorithm:

$$\Pr[\exists i \in [t] \ : \ \langle\mathbf{v}^{(i)}, \mathbf{u}\rangle^2 \ge \tau_i] \le \sum_{i=1}^{t}\Pr[\langle\mathbf{v}^{(i)}, \mathbf{u}\rangle^2 \ge \tau_i \ : \ \forall j \in [i-1] \ \langle\mathbf{v}^{(j)}, \mathbf{u}\rangle^2 \le \tau_j]$$

Now, we bound each summand on the right by using Imported Theorem 34. We take $\theta := \mathbf{u}$, so that $\mathcal{P}$ is the distribution of $\mathbf{u}$. Then, $\mathbb{P}_\theta$ becomes $\mathbb{P}_\mathbf{u}$ from Setting 31, and $\mathbb{Q}$ is exactly $\mathbb{Q}$ as in Setting 31. We take the truncation event $A_\theta = A_\mathbf{u}^{i-1}$. The set of actions $\mathcal{V}$ is conditioned on the prior query vectors $\mathbf{v}^{(1)}, \dots, \mathbf{v}^{(i-1)}$ already made by the algorithm:

$$\mathcal{V}^i := \{\mathbf{v}^{(i)} \ : \ \mathbf{v}^{(i)} = \otimes_{j=1}^{q}\mathbf{v}_j^{(i)}, \ \|\mathbf{v}^{(i)}\|_2 = 1, \ \mathrm{cond}([\,\mathbf{v}^{(1)} \dots \mathbf{v}^{(i)}\,]) \le \kappa\}$$

Lastly, we take $\mathcal{L}(\mathbf{v}^{(i)}, \mathbf{u}) = \mathbb{1}_{[\langle \mathbf{v}^{(i)}, \mathbf{u} \rangle^2 \leq \tau_i]}$. Therefore, Imported Theorem 34 takes

$$V_0 = \sup_{\mathbf{v}^{(i)} \in \mathcal{V}^i} \Pr_{\mathbf{u}} [\langle \mathbf{v}, \mathbf{u} \rangle^2 \geq \tau_i] \leq C_0^{-q}$$

where the last inequality uses Lemma 8, recalling that $\tau_i = C_\tau^{-q}$. So, Imported Theorem 34 tells us that

$$V_v = \mathbb{E}_{\mathbf{u}} \Pr_{\mathcal{Z}_i \sim \mathbb{P}_{\mathbf{u}}} [\mathcal{L}(\mathcal{A}(x), \theta) = 0, A_\theta]$$

$$= \Pr[\langle \mathbf{v}^{(i)}, \mathbf{u} \rangle^2 \geq \tau_i ~:~ \forall j \in [i-1] \langle \mathbf{v}^{(j)}, \mathbf{u} \rangle^2 \leq \tau_i]$$

$$\leq C_0^{-q} + \sqrt{C_0^{-q} \mathbb{E}_{\mathbf{u}} \mathbb{E}_{\mathcal{Z}_i \sim \mathbb{Q}} \left[ \left( \frac{d\mathbb{P}_{\mathbf{u}}[\mathcal{Z}_i]}{d\mathbb{Q}[\mathcal{Z}_i]} \right)^2 \mathbb{1}_{[A_{\mathbf{u}}^{i-1}]} \right]}$$

So, next we bound this expectation. First, we take a moment to examine this ratio of probabilities. We would like to apply Lemma 33 to bound the inner-most expectation. However, the indicator variable inside the expectation prevents us from doing so. Instead, we apply Lemma 49 to this situation (taking $\mathbb{P}_a = \mathbb{P}_b = \mathbb{P}_{\mathbf{u}}$). This lemma tells us that

$$\mathbb{E}_{\mathcal{Z}_i \sim \mathbb{Q}} \left[ \left( \frac{d\mathbb{P}_{\mathbf{u}}[\mathcal{Z}_i]}{d\mathbb{Q}[\mathcal{Z}_i]} \right)^2 \mathbb{1}_{[A_{\mathbf{u}}^{i-1}]} \right] \leq \sup_{\mathcal{Z}_i ~:~ (\mathbf{v}^{(1)}, \cdots, \mathbf{v}^{(i)}) \in A^{i-1}} \prod_{j=1}^{i} \mathbb{E} \left[ \left( \frac{d\mathbb{P}_{\mathbf{h}}(\mathcal{Z}_j \mid \mathcal{Z}_{j-1})}{d\mathbb{Q}(\mathcal{Z}_j \mid \mathcal{Z}_{j-1})} \right)^2 \mid \mathcal{Z}_{j-1} \right].$$

Now, we analyze this conditional expectation on the right. First, recall that we assumed without loss of generality that $\mathcal{A}$ is a determinstic algorithm. Therefore, the $j^{th}$ query vector $\mathbf{v}^{(j)}$ is deterministic in $\mathcal{Z}_{j-1}$. So, the random variable $\mathcal{Z}_j \mid \mathcal{Z}_{j-1}$ is equivalent to just looking at $w_j = \langle \mathbf{v}^{(j)}, \mathbf{a} \rangle$. Formally using the Data Processing Inequality (Lemma E.1 from (Simchowitz et al., 2017)), this means that it suffices to bound

$$\mathbb{E} \left[ \left( \frac{d\mathbb{P}_{\mathbf{h}}(w_j \mid \mathcal{Z}_{j-1})}{d\mathbb{Q}(w_j \mid \mathcal{Z}_{j-1})} \right)^2 \mid \mathcal{Z}_{j-1} \right].$$

Next, we take a moment to analyze the impact of conditioning here. Unfortunately, it is annoying to analyze the expression above due to the way that $w_j = \langle \mathbf{v}^{(j)}, \mathbf{a} \rangle$ may depend on the previous observations in $\mathcal{Z}_{j-1}$. Instead, we appeal to the Data Processing Inequality again to orthonormalize. In particular, for the set of queries $\mathbf{V}_j := [\mathbf{v}^{(1)} ~ \cdots ~ \mathbf{v}^{(j)}]$ in $\mathcal{Z}_j$, we let $\mathbf{X}_j := [\mathbf{x}^{(1)} ~ \cdots ~ \mathbf{x}^{(j)}]$ be the result of running Gram-Schmidt on $\mathbf{V}_j$. That is, $\mathbf{X}_j$ is an orthogonal matrix that spans $\mathbf{V}_j$. We can write our new adjusted transcript as

$$\tilde{\mathcal{Z}}_j = (\mathbf{x}^{(1)}, \langle \mathbf{x}^{(1)}, \mathbf{a} \rangle, \cdots, \mathbf{x}^{(j)}, \langle \mathbf{x}^{(j)}, \mathbf{a} \rangle)$$

Since this process is invertible, it does not change the statistical distance, and therefore it suffices to bound

$$\mathbb{E} \left[ \left( \frac{d\mathbb{P}_{\mathbf{h}}(\langle \mathbf{x}^{(j)}, \mathbf{a} \rangle \mid \tilde{\mathcal{Z}}_{j-1})}{d\mathbb{Q}(\langle \mathbf{x}^{(j)}, \mathbf{a} \rangle \mid \tilde{\mathcal{Z}}_{j-1})} \right)^2 \mid \tilde{\mathcal{Z}}_{j-1} \right].$$

Next, we observe that the set of all observations under the adjusted transcript $\tilde{\mathbf{w}} = [\langle \mathbf{x}^{(1)}, \mathbf{a} \rangle ~ \cdots ~ \langle \mathbf{x}^{(j)}, \mathbf{a} \rangle] = \mathbf{X}_j^\intercal \mathbf{a}$ is distributed as a multivariate Gaussian. Under $\mathbb{Q}$, $\tilde{\mathbf{w}} \sim \mathcal{N}(\mathbf{0}, \mathbf{X}_j^\intercal \mathbf{X}_j) = \mathcal{N}(\mathbf{0}, \mathbf{I})$. Similarly, under $\mathbb{P}_{\mathbf{u}}$, $\tilde{\mathbf{w}} \sim \mathcal{N}(\lambda \mathbf{X}_j^\intercal \mathbf{u}, \mathbf{I})$. Notice that under both distributions, we have that all entries of $\tilde{\mathbf{w}}$ are independent. Therefore, we know that $\langle \mathbf{x}^{(j)}, \mathbf{a} \rangle$ is independent of all other $\langle \mathbf{x}^{(m)}, \mathbf{a} \rangle$ given $\mathbf{X}_j$. So, we can use Lemma 33 to say that

$$\mathbb{E} \left[ \left( \frac{d\mathbb{P}_{\mathbf{h}}(\langle \mathbf{x}^{(j)}, \mathbf{a} \rangle \mid \tilde{\mathcal{Z}}_{j-1})}{d\mathbb{Q}(\langle \mathbf{x}^{(j)}, \mathbf{a} \rangle \mid \tilde{\mathcal{Z}}_{j-1})} \right)^2 \mid \tilde{\mathcal{Z}}_{j-1} \right] = e^{\lambda^2 \langle \mathbf{x}^{(j)}, \mathbf{u} \rangle^2}$$

We want to upper bound this expectation in terms of our original transcript $\mathcal{Z}_t$ though. Here, we use our conditioning assumption. By Lemma 26, we know that $\langle \mathbf{x}^{(j)}, \mathbf{u} \rangle^2 \leq \kappa^2 \|\mathbf{V}_j^\intercal \mathbf{u}\|_2^2$. This means we bound

$$\mathbb{E} \left[ \left( \frac{d\mathbb{P}_{\mathbf{h}}(\langle \mathbf{x}^{(j)}, \mathbf{a} \rangle \mid \tilde{\mathcal{Z}}_{j-1})}{d\mathbb{Q}(\langle \mathbf{x}^{(j)}, \mathbf{a} \rangle \mid \tilde{\mathcal{Z}}_{j-1})} \right)^2 \mid \tilde{\mathcal{Z}}_{j-1} \right] \leq e^{\lambda^2 \kappa^2 \|\mathbf{V}_j^\intercal \mathbf{u}\|_2^2}.$$

Further, using our conditioning assumption, we know that Backing up, we then need to bound

$$
\mathop{\mathbb{E}}_{\mathcal{Z}_i \sim \mathbb{Q}} \left[ \left( \frac{d\mathbb{P}_{\mathbf{u}}[\mathcal{Z}_i]}{d\mathbb{Q}[\mathcal{Z}_i]} \right)^2 \mathbb{1}_{\left[ A_{\mathbf{u}}^{i-1} \right]} \right] \leq \sup_{\mathcal{Z}_i \,:\, (\mathbf{v}^{(1)}, \cdots, \mathbf{v}^{(i)}) \in A^{i-1}} \prod_{j=1}^{i} \mathbb{E} \left[ \left( \frac{d\mathbb{P}_{\mathbf{h}}(\mathcal{Z}_j \mid \mathcal{Z}_{j-1})}{d\mathbb{Q}(\mathcal{Z}_j \mid \mathcal{Z}_{j-1})} \right)^2 \bigg| \mathcal{Z}_{j-1} \right]
$$

$$
= \sup_{\mathcal{Z}_i \,:\, (\mathbf{v}^{(1)}, \cdots, \mathbf{v}^{(i)}) \in A^{i-1}} e^{\lambda^2 \kappa^2 \sum_{j=1}^{i} \| \mathbf{V}_j^{\mathsf{T}} \mathbf{u} \|_2^2}
$$

$$
\leq \sup_{\mathcal{Z}_i \,:\, (\mathbf{v}^{(1)}, \cdots, \mathbf{v}^{(i)}) \in A^{i-1}} e^{\lambda^2 \kappa^2 i \| \mathbf{V}_i^{\mathsf{T}} \mathbf{u} \|_2^2}
$$

$$
\leq e^{\lambda^2 \kappa^2 i \sum_{j=1}^{i} \tau_j}
$$

$$
\leq e^{\lambda^2 \kappa^2 i^2 C_\tau^{-q}}
$$

Then, we can complete our overall lemma by taking

$$
\Pr[\exists i \in [t] \,:\, \langle \mathbf{v}^{(i)}, \mathbf{u} \rangle^2 \geq \tau_i] \leq \sum_{i=1}^{t} \Pr[\langle \mathbf{v}^{(i)}, \mathbf{u} \rangle^2 \geq \tau_i \,:\, \forall j \in [i-1] \, \langle \mathbf{v}^{(j)}, \mathbf{u} \rangle^2 \leq \tau_j]
$$

$$
\leq \sum_{i=1}^{t} \left( C_0^{-q} + \sqrt{C_0^{-q} \mathop{\mathbb{E}}_{\mathbf{u}} \mathop{\mathbb{E}}_{\mathcal{Z}_i \sim \mathbb{Q}} \left[ \left( \frac{d\mathbb{P}_{\mathbf{u}}[\mathcal{Z}_i]}{d\mathbb{Q}[\mathcal{Z}_i]} \right)^2 \mathbb{1}_{\left[ A_{\mathbf{u}}^{i-1} \right]} \right]} \right)
$$

$$
\leq \sum_{i=1}^{t} \left( C_0^{-q} + C_0^{-q/2} e^{\lambda^2 \kappa^2 i^2 C_\tau^{-q}/2} \right)
$$

$$
\leq \sum_{i=1}^{t} 2 C_0^{-q/2} e^{\lambda^2 \kappa^2 i^2 C_\tau^{-q}/2}
$$

$$
\leq 2t C_0^{-q/2} e^{\lambda^2 \kappa^2 t^2 C_\tau^{-q}/2}
$$

$$
\leq \frac{1}{27}
$$

where we take $t = O(\min\{C_0^{q/2}, \frac{C_\tau^{q/2}}{\kappa^2 \lambda}\}) = O(\min\{C_0^{q/2}, \frac{C_\tau^{q/2}}{\kappa^2 \sqrt{\varepsilon}}\})$ on the last line.

$\square$

**Lemma 36.** *Consider Setting 31, where $\tau_1 = \cdots = \tau_t = C_\tau^{-q}$. Then, we have that*

$$
\mathop{\mathbb{E}}_{\mathcal{Z}_t \sim \mathbb{Q}} \left[ \left( \frac{\mathbb{E}_{\mathbf{u}}[d\mathbb{P}_{\mathbf{u}}(\mathcal{Z}_t \cap A_{\mathbf{u}}^t)]}{d\mathbb{Q}(\mathcal{Z}_t)} \right)^2 \right] \leq 1 + z
$$

*so long as $t = O(\min\{C_0^{q/2}, \frac{C_\tau^{q/2}}{\kappa^2 \sqrt{\varepsilon}}\})$.*

*Proof.* Equation (6.31) of (Simchowitz et al., 2017) shows that we can rewrite

$$
\mathop{\mathbb{E}}_{\mathcal{Z}_t \sim \mathbb{Q}} \left[ \left( \frac{\mathbb{E}_{\mathbf{u}}[d\mathbb{P}_{\mathbf{u}}(\mathcal{Z}_t \cap A_{\mathbf{u}}^t)]}{d\mathbb{Q}(\mathcal{Z}_t)} \right)^2 \right] = \mathop{\mathbb{E}}_{\mathbf{u}, \mathbf{u}'} \left[ \mathop{\mathbb{E}}_{\mathcal{Z}_t \sim \mathbb{Q}} \left[ \frac{d\mathbb{P}_{\mathbf{u}}(\mathcal{Z}_t \cap A_{\mathbf{u}}^t) d\mathbb{P}_{\mathbf{u}'}(\mathcal{Z}_t \cap A_{\mathbf{u}'}^t)}{(d\mathbb{Q}(\mathcal{Z}_t))^2} \right] \right]
$$

where $\mathbf{u}'$ is an iid copy of $\mathbf{u}$. Then, by Lemma 49 we know that

$$
\mathop{\mathbb{E}}_{\mathcal{Z}_t \sim \mathbb{Q}} \left[ \frac{d\mathbb{P}_{\mathbf{u}}(\mathcal{Z}_t \cap A_{\mathbf{u}}^t) d\mathbb{P}_{\mathbf{u}'}(\mathcal{Z}_t \cap A_{\mathbf{u}'}^t)}{(d\mathbb{Q}(\mathcal{Z}_t))^2} \right] \leq \sup_{\mathcal{Z}_t \in A_{\mathbf{u}}^t \cap A_{\mathbf{u}'}^t} \prod_{i=1}^{t} \mathbb{E} \left[ \frac{d\mathbb{P}_{\mathbf{u}}(\mathcal{Z}_i \mid \mathcal{Z}_{i-1}) d\mathbb{P}_{\mathbf{u}'}(\mathcal{Z}_i \mid \mathcal{Z}_{i-1})}{(d\mathbb{Q}(\mathcal{Z}_i \mid \mathcal{Z}_{i-1}))^2} \big| \mathcal{Z}_{i-1} \right].
$$

As in Lemma 35, we change our basis from $\mathbf{V}_j$ to $\mathbf{X}_j$. Under $\mathbb{Q}$, we have $\tilde{\mathbf{w}} \sim \mathcal{N}(\mathbf{0}, \mathbf{I})$. Under $\mathbb{P}_{vu}$, we have $\tilde{\mathbf{w}} \sim \mathcal{N}(\lambda \mathbf{X}_j^\intercal \mathbf{u}, \mathbf{I})$. So, by Lemma 33, we know that

$$\mathbb{E}\left[\frac{d\mathbb{P}_\mathbf{u}(\mathcal{Z}_i \mid \mathcal{Z}_{i-1})d\mathbb{P}_{\mathbf{u}'}(\mathcal{Z}_i \mid \mathcal{Z}_{i-1})}{(d\mathbb{Q}(\mathcal{Z}_i \mid \mathcal{Z}_{i-1}))^2}\Big| \mathcal{Z}_{i-1}\right] = e^{\lambda^2 \langle \mathbf{x}^{(j)}, \mathbf{u}\rangle\langle \mathbf{x}^{(j)}, \mathbf{u}'\rangle}$$

And, again following Lemma 26, we bound $|\langle \mathbf{x}^{(j)}, \mathbf{u}\rangle| \leq \kappa \|\mathbf{V}_j^\intercal \mathbf{u}\|_2$, so the above exponential is at most $e^{\lambda^2 \kappa^2 \|\mathbf{V}_j^\intercal \mathbf{u}\|_2^2}$. Therefore,

$$\begin{aligned}
\mathbb{E}_{\mathcal{Z}_t \sim \mathbb{Q}}\left[\frac{d\mathbb{P}_\mathbf{u}(\mathcal{Z}_t \cap A_\mathbf{u}^t)d\mathbb{P}_{\mathbf{u}'}(\mathcal{Z}_t \cap A_{\mathbf{u}'}^t)}{(d\mathbb{Q}(\mathcal{Z}_t))^2}\right] &\leq \sup_{\mathcal{Z}_t \in A_\mathbf{u}^t \cap A_{\mathbf{u}'}^t} \prod_{i=1}^t \mathbb{E}\left[\frac{d\mathbb{P}_\mathbf{u}(\mathcal{Z}_i \mid \mathcal{Z}_{i-1})d\mathbb{P}_{\mathbf{u}'}(\mathcal{Z}_i \mid \mathcal{Z}_{i-1})}{(d\mathbb{Q}(\mathcal{Z}_i \mid \mathcal{Z}_{i-1}))^2}\Big| \mathcal{Z}_{i-1}\right] \\
&\leq \sup_{\mathcal{Z}_t \in A_\mathbf{u}^t \cap A_{\mathbf{u}'}^t} e^{\lambda^2 \kappa^2 \sum_{i=1}^t \|\mathbf{V}_j^\intercal \mathbf{u}\|_2^2} \\
&\leq \sup_{\mathcal{Z}_t \in A_\mathbf{u}^t \cap A_{\mathbf{u}'}^t} e^{\lambda^2 \kappa^2 t \|\mathbf{V}_t^\intercal \mathbf{u}\|_2^2} \\
&\leq e^{\lambda^2 \kappa^2 t \sum_{i=1}^t \tau_i} \\
&\leq e^{\lambda^2 \kappa^2 t^2 C_\tau^{-q}} \\
&\leq 1 + \frac{1}{27}
\end{aligned}$$

where we take $t = O(\min\{C_0^{q/2}, \frac{C_\tau^{q/2}}{\kappa^2 \lambda}\}) = O(\min\{C_0^{q/2}, \frac{C_\tau^{q/2}}{\kappa^2 \sqrt{\varepsilon}}\})$ on the last line, completing the proof. □

# C. Formal Adaptive Matrix-Vector Lower Bound

Written in the notation and form of Appendix D, we can write Theorem 15 equivalently as the following:

**Theorem 15 Restated.** *Consider Setting 39 where $\mathcal{V}^t$ is the set of $\kappa-$conditioned Kronecker-structured query vectors:*

$$\mathcal{V}^t := \{(\mathbf{v}^{(1)}, \dots, \mathbf{v}^{(t)}) \ : \ \mathbf{v}^{(i)} = \otimes_{j=1}^q \mathbf{v}_j^{(i)}, \ \mathbf{v}_j^{(i)} \in \mathbb{S}^n, \ cond([\mathbf{v}^{(1)} \ \cdots \ \mathbf{v}^{(t)}]) \leq \kappa\}$$

*Then, $t = \Omega(\min\{C_0^{q/2}, \frac{C_\tau^q}{\lambda^2 \kappa^2}\})$ matrix-vector products are needed to correctly guess if $\mathbf{A} = \mathbf{A}_0$ or $\mathbf{A}_1$ in Setting 39 with probability at least $\frac{2}{3}$, where $C_0$ and $C_\tau$ are the constants in Lemma 8.*

*Proof.* We proceed by using Theorem 40 in conjunction with Lemma 8 and Lemma 25. In particular, we Lemma 8 tells us that $f(\tau) \leq C_0^{-q}$ for any $\tau \leq C_\tau^{-q}$. Therefore, we can take $\tau_1 = \dots = \tau_t = C_\tau^{-q}$ so that $\sum_{j=1}^i \tau_j = iC_\tau^q$. By assuming that $t \leq \frac{3C_\tau^q}{\lambda^2 \kappa^2}$, we know that $e^{\frac{\lambda^2 \kappa^2}{2} \sum_{j=1}^{i=1} \tau_j} \leq e^{\frac{\lambda^2 \kappa^2 i}{2C_\tau^q}} \leq 4$. Then,

$$\begin{aligned}
f(\tau_1) + 2\sum_{i=1}^t e^{\frac{\lambda^2 \kappa^2}{2} \sum_{j=1}^{i=1} \tau_j} \sqrt{f(\tau_j)} &\leq C_0^{-q} + 8\sum_{i=1}^t C_0^{-q/2} \\
&\leq (1 + 8t)C_0^{-q/2} \\
&\leq \frac{1}{27}
\end{aligned}$$

where the last line holds so long as $t \leq O(C_0^{-q/2})$. Similarly, we can use Lemma 25 to bound $\mathbb{E}_{\mathbf{u},\mathbf{u}'}[e^{\eta|\langle \mathbf{u},\mathbf{u}'\rangle|}] \leq e^{\frac{2\eta}{n^q}}$ for

any $\eta \in (0,1)$. Therefore,

$$
\mathbb{E}_{\mathbf{u},\mathbf{u}'\sim\mathcal{P}}[e^{\lambda^2\kappa^2|\langle\mathbf{u},\mathbf{u}'\rangle|(\sum_{i=1}^{t}\tau_i)+\frac{\lambda^2\kappa^4}{n^q}(\sum_{i=1}^{t}\tau_i)^2}] = \mathbb{E}_{\mathbf{u},\mathbf{u}'\sim\mathcal{P}}\left[e^{\frac{\lambda^2\kappa^2 t}{C_\tau^q}|\langle\mathbf{u},\mathbf{u}'\rangle|}\right]e^{\frac{\lambda^2\kappa^4 t^2}{n^q C_\tau^{2q}}}
$$

$$
\leq e^{2\frac{\lambda^2\kappa^2 t}{n^q C_\tau^q}+\frac{\lambda^2\kappa^4 t^2}{n^q C_\tau^{2q}}}
$$

$$
= e^{2\frac{\lambda^2\kappa^2 t}{n^q C_\tau^q}(1+\frac{\kappa^2 t}{4C_\tau^q})}
$$

$$
\leq e^{4\frac{\lambda^2\kappa^2 t}{n^q C_\tau^q}}
$$

$$
\leq 1+8\frac{\lambda^2\kappa^2 t}{n^q C_\tau^q}
$$

$$
\leq 1+\frac{1}{27}
$$

where we use that $\eta = \frac{\lambda^2\kappa^2 t}{C_\tau^q} \leq 1$ in the first inequality, that $t \leq \frac{4C_\tau^q}{\kappa^2}$ in the second inequality, that $e^x \leq 1+2x$ for $x \leq 1$ in the third inequality, and we take $t \leq \frac{n^q C_\tau^q}{27\cdot 8\lambda^2\kappa^2}$ in the last inequality. By Theorem 40, we find that having $t \leq O(\min\{C_0^{q/2}, \frac{C_\tau^q}{\lambda^2\kappa^2}\})$ does not suffice to correctly guess if $\mathbf{A} = \mathbf{A}_0$ or $\mathbf{A} = \mathbf{A}_1$ in Setting 39 with probability at least $\frac{2}{3}$. □

## D. Connecting to the Simchowitz et. al Lower Bounds

In (Simchowitz et al., 2017), the authors prove a lower bound against the number of matrix-vector products needed to detect if there is a rank-one matrix planted on a random Wigner matrix. Their techniques and proofs are all written to consider the generic matrix-vector model, where we can compute $\mathbf{A}\mathbf{v}$ for any vector $\mathbf{v} \in \mathbb{R}^D$. However, with minor alteration, their proof techniques can be significantly generalized to allow matrix-vector products with a limited matrix-vector model. To start, we define a generic notion of a limited matrix-vector model.

**Definition 37.** *Fix a set $\mathcal{V}^t \in (\mathbb{S}^D)^t$. A matrix-vector algorithm $\mathcal{A}$ is $\mathcal{V}^t$ limited if it always computed exactly $t$ matrix vector products and if, for all input matrices $\mathbf{A} \in \mathbb{R}^{D\times D}$ the algorithm only computes (possibly adaptive) query vectors $\mathbf{v}^{(1)},\ldots,\mathbf{v}^{(t)}$ such that the sequence $(\mathbf{v}^{(1)},\ldots,\mathbf{v}^{(t)}) \in \mathcal{V}^t$.*

The proof methods of (Simchowitz et al., 2017) rely on assuming that the matrix-vector queries computed are orthonormal. We do not want to assume that the queries admissible in $\mathcal{V}^t$ are orthonormal, so we instead will make an assumption on $\mathcal{V}^t$ that measure how far $\mathcal{V}^t$ is from having orthonormal queries:

**Definition 38.** *For each $(\mathbf{v}^{(1)},\ldots,\mathbf{v}^{(t)}) \in \mathcal{V}^t$, let $\mathbf{V} = [\mathbf{v}^{(1)} \ldots \mathbf{v}^{(t)}] \in \mathbb{R}^{D\times t}$. If, for all $(\mathbf{v}^{(1)},\ldots,\mathbf{v}^{(t)}) \in \mathcal{V}^t$ we know that the condition number of $\mathbf{V}$ is at most $\kappa$, then we say that $\mathcal{V}^t$ is $\kappa-$conditioned.*

We can now setup the instance of the lower bound problem considered in (Simchowitz et al., 2017).

**Setting 39.** *Fix $D \in \mathbb{N}$ and $\lambda > 1$. Fix a $\mathcal{V}^t$ limited matrix-vector algorithm $\mathcal{A}$. Let $\mathcal{P}$ be a isotropic prior distribution over planted vectors $\mathbf{u} \in \mathbb{S}^{D-1}$, so that $\mathbb{E}[\mathbf{u}] = \mathbf{0}$ and $\mathbb{E}[\mathbf{u}\mathbf{u}^\intercal] = \mathbf{I}$. Let $\mathbf{W} = \frac{1}{2}(\mathbf{G} + \mathbf{G}^\intercal)$ where $\mathbf{G} \in \mathbb{R}^{D\times D}$ is a matrix with iid $\mathcal{N}(0,1)$ entries. Let $\mathbf{A}_0 := \frac{1}{\sqrt{D}}\mathbf{W}$ and $\mathbf{A}_1 := \frac{1}{\sqrt{D}}\mathbf{W} + \lambda\mathbf{u}\mathbf{u}^\intercal$. Nature picks $i \in \{0,1\}$ uniformly at random. $\mathcal{A}$ then computes $t$ matrix vector products with $\mathbf{A} := \mathbf{A}_i$ and returns a guess of the value of $i$.*

In this setting, we will show that (Simchowitz et al., 2017) proves the following lower bounding mechanism:

**Theorem 40.** *Consider Setting 39. Fix any $0 \leq \tau_1 \leq \ldots \leq \tau_t$ and $\kappa > 1$. Let $f(\tau)$ be the probability that the best possible blind guess for $\mathbf{u}$ in $\mathcal{V}^t$ has squared inner product at least $\frac{\tau}{D}$:*

$$
f(\tau) := \sup_{(\mathbf{v}^{(1)},\ldots,\mathbf{v}^{(t)})\in\mathcal{V}^t} \sup_{i\in[t]} \Pr[\langle\mathbf{v}^{(i)},\mathbf{u}\rangle^2 > \tfrac{\tau}{D}].
$$

*Suppose that for some $z \in (0,1)$*

$$
f(\tau_1) + 2\sum_{i=1}^{t} e^{\frac{\lambda^2\kappa}{2}\sum_{j=1}^{i-1}\tau_j}\sqrt{f(\tau_j)} \leq z
$$

*and that*

$$\mathop{\mathbb{E}}_{\mathbf{u},\mathbf{u}'\sim\mathcal{P}}[e^{\lambda^2\kappa^2|\langle\mathbf{u},\mathbf{u}'\rangle|\sum_{i=1}^t\tau_i+\frac{\lambda^2\kappa^4}{D}(\sum_{i=1}^t\tau_i)^2}]\le 1+z.$$

*Then, $\mathcal{A}$ can distinguish $\mathbf{A}_0$ from $\mathbf{A}_1$ with probability at most $\frac{1}{2}+\frac{1}{2}\sqrt{3z}$. In particular, if $z\le\frac{1}{27}$, then any such algorithm $\mathcal{A}$ cannot correctly guess if $i=0$ or $i=1$ with probability at least $\frac{2}{3}$.*

In Appendix C, we show how to use Theorem 40 to lower bound Kronecker matrix-vector complexity. In this section, we instead will show how Theorem 40 follow from (Simchowitz et al., 2017). To start, we will use the notion of *Truncated Probability Distributions* from (Simchowitz et al., 2017).

**Definition 41.** *Let $\mathbb{P}$ be a probability measure. Let $A$ be an event. Then, the truncated probability measure of $\mathbb{P}$ with respect to $A$ is defined by saying for all events $B$,*

$$P[B;A] := P[B\cap A]$$

This is not a probability distribution as its integral is less than 1 for any nontrivial event $A$. For discussion as to why the truncated probability distribution is helpful in proving information-theoretic lower bounds, see (Simchowitz et al., 2017; 2018). We will also need the idea of the marginal of truncated distributions.

**Definition 42.** *Let $\mathcal{P}$ be a distribution over a space $\Theta$. Let $\{\mathbb{P}_\theta\}_{\theta\in\Theta}$ be a family of probability measures on space $(\mathcal{X},\mathcal{F})$. For each $\theta$, let $A_\theta$ be an event on $\mathcal{F}$. For any event $B$ on $\mathcal{F}$ we can then define the* marginal truncated distribution $\bar{\mathbb{P}}[\cdot;\bar{A}]$ *as*

$$\bar{\mathbb{P}}[B;\bar{A}] := \mathop{\mathbb{E}}_{\theta\sim\mathcal{P}}\mathbb{P}_\theta[B;A_\theta].$$

*Notice that the total measure of $\bar{\mathbb{P}}[\cdot,\bar{A}]$ is $\bar{\mathbb{P}}[\mathcal{X};\bar{A}] = \mathbb{E}_{\theta\sim\mathcal{P}}\mathbb{P}_\theta[\mathcal{X};A_\theta] = \mathrm{Pr}_{\theta\sim\mathcal{P}}[A_\theta]$. Without truncation, we write $\bar{\mathbb{P}} := \bar{\mathbb{P}}[\cdot;\mathcal{X}]$.*

We will concretely take $\mathbb{Q}$ and $\bar{\mathbb{P}}$ to be distributions over transcripts of matrix-vector products between $\mathcal{A}$ and $\mathbf{A}$. That is, we let $\mathcal{Z}_t := (\mathbf{v}^{(1)},\mathbf{A}\mathbf{v}^{(1)},\ldots,\mathbf{v}^{(t)},\mathbf{A}\mathbf{v}^{(t)})$ be the transcript of $t$ matrix-vector products. Then, we take $\mathbb{Q}$ to be the distribution of $\mathcal{Z}_t$ given $i=0$, so that $\mathbf{A} = \frac{1}{\sqrt{D}}\mathbf{W}$. We then will take $\mathbf{u}\sim\mathcal{P}$ as our prior distribution, so that $\theta=\mathbf{u}$. Then, we let $\mathbb{P}_\mathbf{u}$ be the distribution of $\mathcal{Z}_t$ given both $i=1$ and a fixed value of $\mathbf{u}$, so that $\mathbf{A} = \frac{1}{\sqrt{D}}\mathbf{W} + \lambda\mathbf{u}\mathbf{u}^\intercal$ for a fixed $\mathbf{u}$. For any fixed $\mathbf{u}$, we will take our truncation event to be $A_\theta = \mathcal{V}_\mathbf{u}^t := \{(\mathbf{v}^{(1)},\ldots,\mathbf{v}^{(t)})\in\mathcal{V}^t : \langle\mathbf{v}^{(i)},\mathbf{u}\rangle^2\le\frac{\tau_i}{D}\,\forall i\in[t]\}$, using the constants $0\le\tau_1\le\ldots\le\tau_t$ as given in Theorem 40. In words, this truncation set $\mathcal{V}_\mathbf{u}^t$ is the set of all queries that fail to find nontrivial information about the vector $\mathbf{u}$. Lastly, this means that $\bar{\mathbb{P}}$ is the marginal distribution of all the truncated distributions. That is, $\bar{\mathbb{P}}$ is the distribution of $\mathcal{Z}_t$ given $i=1$ but not given any particular value of $\mathbf{u}$, and $\bar{\mathbb{P}}[\cdot,\bar{A}]$ is $\bar{\mathbb{P}}$ truncated to the cases where our algorithm has not computed any matrix-vector products that achieve nontrivial inner product with $\mathbf{u}$.

Our main goal is to bound the total variation distance between $\mathbb{Q}$ and $\bar{\mathbb{P}}$. (Simchowitz et al., 2017) bound this distance by truncating $\bar{\mathbb{P}}$ and bounding both the probability of the truncated event not happening and the distance between $\mathbb{Q}$ and the truncated $\bar{\mathbb{P}}[\cdot;\bar{A}]$. This is formalized by Proposition 6.1 from (Simchowitz et al., 2017), whose proof has a fixable error. We provide and prove the fixed version below:

**Imported Lemma 43** (Proposition 6.1 of (Simchowitz et al., 2017)). *Let $\mathcal{P}$, $\{\mathbb{P}_\theta\}_{\theta\in\Theta}$, and $A_\theta$ define a marginal truncated distribution $\bar{\mathbb{P}}[\cdot,\bar{A}]$ on $(\mathcal{X},\mathcal{F})$. Let $\mathbb{Q}$ be a probability distribution on $(\mathcal{X},\mathcal{F})$. Then, letting $p := \bar{\mathbb{P}}[\mathcal{X};\bar{A}] = \mathrm{Pr}_{\theta\sim\mathcal{P}}[A_\theta]$, we have*

$$D_{TV}(\bar{\mathbb{P}},\mathbb{Q})\le\frac{1}{2}\sqrt{\mathop{\mathbb{E}}_{x\sim\mathbb{Q}}\left[\left(\frac{d\bar{\mathbb{P}}[x;\bar{A}]}{d\mathbb{Q}(x)}\right)^2\right]+1-2p}+\frac{1-p}{2}.$$

*In particular, if we have $\mathbb{E}_{x\sim\mathbb{Q}}\left[\left(\frac{d\bar{\mathbb{P}}[x;\bar{A}]}{d\mathbb{Q}(x)}\right)^2\right]\le 1+z$ and $1-p<z$ for some $z\in(0,1)$, then we can bound $D_{TV}(Q,\bar{P})\le\sqrt{3z}$.*

*Proof. This proof is a close copy of the result in (Simchowitz et al., 2017), but avoids errors in algebra. For ease of notation, let $\bar{\mathbb{P}}_A := \bar{\mathbb{P}}[\cdot,\bar{A}]$. Note that $\bar{\mathbb{P}}-\bar{\mathbb{P}}_A\ge 0$, which implies that*

$$\int|d\bar{\mathbb{P}}-d\bar{\mathbb{P}}_A| = \int d\bar{\mathbb{P}}-d\bar{\mathbb{P}}_A = 1-p$$

so by the triangle inequality,

$$D_{TV}(\mathbb{Q}, \bar{\mathbb{P}}) = \frac{1}{2} \int |d\mathbb{Q}(x) - d\bar{\mathbb{P}}(x)|$$
$$\leq \frac{1}{2} \int |d\mathbb{Q}(x) - d\bar{\mathbb{P}}_A(x)| + \frac{1}{2} \int |d\bar{\mathbb{P}}(x) - d\bar{\mathbb{P}}_A(x)|$$
$$= \frac{1}{2} \int |d\mathbb{Q}(x) - d\bar{\mathbb{P}}_A(x)| + \frac{1-p}{2}.$$

Next, since $\mathbb{Q}$ is a probability measure,

$$\int |d\mathbb{Q}(x) - d\bar{\mathbb{P}}_A(x)| = \mathop{\mathbb{E}}_{\mathbb{Q}} |\tfrac{d\bar{\mathbb{P}}_A(x)}{d\mathbb{Q}(x)} - 1| \leq \sqrt{\mathop{\mathbb{E}}_{\mathbb{Q}} |\tfrac{d\bar{\mathbb{P}}_A(x)}{d\mathbb{Q}(x)} - 1|^2}$$
$$= \sqrt{\mathop{\mathbb{E}}_{\mathbb{Q}} |\tfrac{d\bar{\mathbb{P}}_A(x)}{d\mathbb{Q}(x)}|^2 + 1 - 2\bar{P}_A(\mathcal{X})} = \sqrt{\mathop{\mathbb{E}}_{\mathbb{Q}} |\tfrac{d\bar{\mathbb{P}}_A(x)}{d\mathbb{Q}(x)}|^2 + 1 - 2p}.$$

Combining what we've shown, we conclude the first result, that

$$D_{TV}(\mathbb{Q}, \bar{\mathbb{P}}) \leq \frac{1}{2} \sqrt{\mathop{\mathbb{E}}_{\mathbb{Q}} \left| \frac{d\bar{\mathbb{P}}_A(x)}{d\mathbb{Q}(x)} \right|^2 + 1 - 2p} + \frac{1-p}{2}.$$

Now, we move onto the second result. Directly substituting our values for $z$ and $1+z$, we get

$$D_{TV}(\mathbb{Q}, \bar{\mathbb{P}}) \leq \frac{1}{2}\sqrt{1 + z + 1 - 2p} + \frac{1-p}{2}$$
$$= \frac{1}{2}\sqrt{z + 2(1-p)} + \frac{1-p}{2}$$
$$\leq \frac{1}{2}\sqrt{z + 2z} + \frac{z}{2}$$
$$\leq \frac{1}{2}\sqrt{3z} + \frac{1}{2}\sqrt{z}$$
$$\leq \sqrt{3z}$$

$\square$

We next import the results that (Simchowitz et al., 2017) used to bound $\mathbb{E}_{\mathcal{Z}_t \sim \mathbb{Q}} \left[ \left( \frac{d\mathbb{P}[\mathcal{Z}_t; \mathcal{V}_{\mathbf{u}}^t]}{d\mathbb{Q}(\mathcal{Z}_t)} \right)^2 \right]$ and $1 - p = 1 - \Pr[\mathcal{V}_{\mathbf{u}}^t] = \Pr[\exists i \in [t] \ : \ \langle \mathbf{v}^{(i)}, \mathbf{u} \rangle^2 > \frac{\tau_i}{D}]$. Starting with $1 - p$, we import Theorem 5.3:

**Imported Theorem 44** (Theorem 5.3 from (Simchowitz et al., 2017)). *Consider Setting 39. Fix $0 \leq \tau_1 \leq \ldots \leq \tau_t$. Let $f(\tau)$ be the probability that the best possible blind guess for $\mathbf{u}$ using a vector in $\mathcal{V}^t$ achieves squared inner product at least $\frac{\tau}{D}$:*

$$f(\tau) := \sup_{(\mathbf{v}^{(1)}, \ldots, \mathbf{v}^{(t)}) \in \mathcal{V}^t} \sup_{i \in [t]} \Pr[\langle \mathbf{v}^{(i)}, \mathbf{u} \rangle^2 > \frac{\tau}{D}].$$

*Then, $\Pr[\exists i \in [t] \ : \ \langle \mathbf{v}^{(i)}, \mathbf{u} \rangle^2 > \frac{\tau_i}{D}]$ is at most*

$$f(\tau_1) + 2 \sum_{i=1}^{t} \mathop{\mathbb{E}}_{\mathbf{u} \sim \mathcal{P}} \left[ \sqrt{f(\tau_i) \sup_{(\tilde{\mathbf{v}}^{(1)}, \ldots, \tilde{\mathbf{v}}^{(t)}) \in \mathcal{V}_{\mathbf{u}}^t} \prod_{j=1}^{i} \mathbb{E} \left[ \left( \frac{d\mathbb{P}_{\mathbf{u}}(\mathbf{A}\tilde{\mathbf{v}}^{(j)} | \tilde{\mathbf{v}}^{(1)}, \ldots, \tilde{\mathbf{v}}^{(t)})}{d\mathbb{Q}(\mathbf{A}\tilde{\mathbf{v}}^{(j)} | \tilde{\mathbf{v}}^{(1)}, \ldots, \tilde{\mathbf{v}}^{(t)})} \right)^2 \ | \ \tilde{\mathbf{v}}^{(1)}, \ldots, \tilde{\mathbf{v}}^{(t)} \right]} \right]$$

This theorem exactly matches Theorem 5.3 from (Simchowitz et al., 2017) if we make two changes. First, we do not yet apply the inequality (5.25) for reasons we will discuss momentarily. Second, we change their set $\mathcal{V}_\theta^k$ into our set $\mathcal{V}_{\mathbf{u}}^t$ in Lemma 5.2 so that we only consider queries that belong to $\mathcal{V}^t$ as opposed to arbitrary query vectors in $\mathbb{S}^D$. The proof of Lemma 5.2 does not change from this redefinition of $\mathcal{V}_{\mathbf{u}}^t$. Next, we must bound this big expectation that appears on the

right hand side above. In inequality (5.25), (Simchowitz et al., 2017) bounds this expectation under the assumption that $\mathbf{v}^{(1)}, \ldots, \mathbf{v}^{(t)}$ are orthonormal. We cannot make this assumption, as an orthonormal basis for vectors in $\mathcal{V}^t$ might not belong to $\mathcal{V}^t$. So, we rephrase Lemma C.3 from (Simchowitz et al., 2017), making this orthonormality reduction explicit:

**Imported Lemma 45** (Lemma C.3 from (Simchowitz et al., 2017)). *Consider Imported Theorem 44. Let $\tilde{\mathbf{x}}^{(1)}, \ldots, \tilde{\mathbf{x}}^{(t)}$ be orthonormal vectors such that $span\{\tilde{\mathbf{x}}^{(1)}, \ldots, \tilde{\mathbf{x}}^{(i)}\} = span\{\tilde{\mathbf{v}}^{(1)}, \ldots, \tilde{\mathbf{v}}^{(i)}\}$ for all $i \in [t]$. Then,*

$$\mathbb{E}\left[\left(\frac{d\mathbb{P}_{\mathbf{u}}(\mathbf{A}\tilde{\mathbf{v}}^{(j)}|\tilde{\mathbf{v}}^{(1)}, \ldots, \tilde{\mathbf{v}}^{(t)})}{d\mathbb{Q}(\mathbf{A}\tilde{\mathbf{v}}^{(j)}|\tilde{\mathbf{v}}^{(1)}, \ldots, \tilde{\mathbf{v}}^{(t)})}\right)^2 \mid \tilde{\mathbf{v}}^{(1)}, \ldots, \tilde{\mathbf{v}}^{(t)}\right] \leq e^{\lambda^2 D \langle \mathbf{u}, \tilde{\mathbf{x}}^{(i)} \rangle^2}$$

Applying this result to Imported Theorem 44, we see that we need to upper bound the expression

$$\sup_{(\tilde{\mathbf{v}}^{(1)}, \ldots, \tilde{\mathbf{v}}^{(t)}) \in \mathcal{V}_{\mathbf{u}}^t} e^{\lambda^2 D \langle \mathbf{u}, \tilde{\mathbf{x}}^{(i)} \rangle^2}$$

Unfortunately, it is not immediately obvious how to relate $\langle \mathbf{u}, \tilde{\mathbf{x}}^{(i)} \rangle$ to $\tilde{\mathbf{v}}^{(1)}, \ldots \tilde{\mathbf{v}}^{(i)}$ or $\tau_1, \ldots, \tau_i$. In the non-Kronecker case, when $\mathcal{V}^t$ covers all vectors in $\mathbb{S}^D$, we can take $\tilde{\mathbf{x}}^{(i)} = \tilde{\mathbf{v}}^{(i)}$ without loss of generality. However, if $\mathcal{V}^t$ covers Kronecker-structured query vectors, then we do not know what the worst-case relationship between these terms is. So, we make an assumption on $\mathcal{V}^t$ to proceed. In particular, we assume that $\mathcal{V}^t$ is well conditioned.

**Lemma 46.** *Consider Imported Theorem 44. Under the assumption that $\mathcal{V}^t$ is $\kappa$-conditioned, we know that*

$$\Pr[\exists i \in [t] \ : \ \langle \mathbf{v}^{(i)}, \mathbf{u} \rangle^2 > \tfrac{\tau_i}{D}] \leq f(\tau_1) + 2\sum_{i=1}^{t} e^{\frac{\lambda^2 \kappa^2}{2}\kappa \sum_{j=1}^{i-1} \tau_j} \sqrt{f(\tau_i)}$$

*Proof.* From the definition of the conditioning of $\mathcal{V}^t$, we know that for all $(\tilde{\mathbf{v}}^{(1)}, \ldots, \tilde{\mathbf{v}}^{(t)}) \in \mathcal{V}_{\mathbf{u}}^t$ that $\tilde{\mathbf{V}} := \begin{bmatrix} \tilde{\mathbf{v}}^{(1)} & \cdots & \tilde{\mathbf{v}}^{(t)} \end{bmatrix}$ has condition number at most $\kappa$. Therefore, by Lemma 26, we know that $\langle \mathbf{u}, \tilde{\mathbf{x}}^{(i)} \rangle^2 \leq \kappa^2 \sum_{j=1}^{i} \langle \mathbf{u}, \tilde{\mathbf{v}}^{(j)} \rangle^2$. By our definition of $\mathcal{V}_{\mathbf{u}}^t$, we further know that $\langle \mathbf{u}, \tilde{\mathbf{v}}^{(j)} \rangle^2 \leq \frac{\tau_j}{D}$. So, we get that

$$\sup_{(\tilde{\mathbf{v}}^{(1)}, \ldots, \tilde{\mathbf{v}}^{(t)}) \in \mathcal{V}_{\mathbf{u}}^t} e^{\frac{\lambda^2}{2} D \langle \mathbf{u}, \tilde{\mathbf{x}}^{(i)} \rangle^2} \sqrt{f(\tau_i)} \leq e^{\frac{\lambda^2}{2}\kappa^2 \sum_{j=1}^{i} \tau_j} \sqrt{f(\tau_i)}$$

Overall, going back to Imported Theorem 44, we find that

$$\Pr[\exists i \in [t] \ : \ \langle \mathbf{v}^{(i)}, \mathbf{u} \rangle^2 > \tfrac{\tau_i}{D}] \leq f(\tau_1) + 2\sum_{i=1}^{t} e^{\frac{\lambda^2 \kappa^2}{2} \sum_{j=1}^{i} \tau_j} \sqrt{f(\tau_i)},$$

completing the proof. $\qquad\square$

This suffices to bound the term $1 - p$ in Imported Lemma 43. However, we still have do bound the expected squared likelihood ratio term. Lemma C.3 from (Simchowitz et al., 2017) is analogous to Imported Lemma 45 but instead applies to this context:

**Imported Lemma 47** (Lemma C.3 from (Simchowitz et al., 2017)). *Consider Imported Theorem 44. Let $\bar{\bar{\mathbb{P}}}$ be the distribution of $\mathcal{Z}_t$ conditioned on $i = 1$ but not conditioned on a specific value of $\mathbf{u} \sim \mathcal{P}$. Let $\tilde{\mathbf{x}}^{(1)}, \ldots, \tilde{\mathbf{x}}^{(t)}$ be orthonormal vectors such that $span\{\tilde{\mathbf{x}}^{(1)}, \ldots, \tilde{\mathbf{x}}^{(i)}\} = span\{\tilde{\mathbf{v}}^{(1)}, \ldots, \tilde{\mathbf{v}}^{(i)}\}$ for all $i \in [t]$. Then,*

$$\mathbb{E}_{\mathcal{Z}_t \sim \mathbb{Q}}\left[\left(\frac{d\bar{\bar{\mathbb{P}}}[\mathcal{Z}_t; \mathcal{V}_{\mathbf{u}}^t]}{d\mathbb{Q}(\mathcal{Z}_t)}\right)^2\right]$$

$$\leq \mathbb{E}_{\mathbf{u}, \mathbf{u}' \sim \mathcal{P}}\left[\sup_{(\tilde{\mathbf{v}}^{(1)}, \ldots, \tilde{\mathbf{v}}^{(t)}) \in \mathcal{V}_{\mathbf{u}}^t \cap \mathcal{V}_{\mathbf{u}'}^t} e^{D\lambda^2 \sum_{i=1}^{t} \langle \tilde{\mathbf{x}}^{(i)}, \mathbf{u} \rangle \langle \tilde{\mathbf{x}}^{(i)}, \mathbf{u}' \rangle \left(\langle \mathbf{u}, \mathbf{u}' \rangle - \frac{1}{2}\langle \tilde{\mathbf{x}}^{(i)}, \mathbf{u} \rangle \langle \tilde{\mathbf{x}}^{(i)}, \mathbf{u}' \rangle - \sum_{j=1}^{i-1} \langle \tilde{\mathbf{x}}^{(j)}, \mathbf{u} \rangle \langle \tilde{\mathbf{x}}^{(j)}, \mathbf{u}' \rangle\right)}\right]$$

Again, Lemma C.4 is not phrased exactly this way in (Simchowitz et al., 2017). This result follows from the proof of Lemma C.4 without substituting the final inequality on page 30. In order to resolve the impact of orthonormality on this proof, we again appeal to conditioning:

**Lemma 48.** *Consider Imported Lemma 47. Under the assumption that $\mathcal{V}^t$ is $\kappa-$conditioned, we know that*

$$\underset{\mathcal{Z}_t \sim \mathbb{Q}}{\mathbb{E}} \left[ \left( \frac{d\bar{\mathbb{P}}[\mathcal{Z}_t; \mathcal{V}_{\mathbf{u}}^t]}{d\mathbb{Q}(\mathcal{Z}_t)} \right)^2 \right] \leq \underset{\mathbf{u},\mathbf{u}'\sim\mathcal{P}}{\mathbb{E}} \left[ e^{\lambda^2 \kappa^2 \langle \tilde{\mathbf{u}}, \mathbf{u}' \rangle \sum_{i=1}^{t} \tau_i + \frac{\lambda^2 \kappa^4}{D} (\sum_{i=1}^{t} \tau_i)^2} \right]$$

*Proof.* As in the proof of Lemma 46, we know that $|\langle \tilde{\mathbf{x}}^{(i)}, \mathbf{u} \rangle| \leq \kappa \frac{\sqrt{\tau_i}}{\sqrt{D}}$ and $|\langle \tilde{\mathbf{x}}^{(i)}, \mathbf{u}' \rangle| \leq \kappa \frac{\sqrt{\tau_i}}{\sqrt{D}}$. So, directly bounding the terms in Imported Lemma 47,

$$D \sum_{i=1}^{t} \langle \tilde{\mathbf{x}}^{(i)}, \mathbf{u} \rangle \langle \tilde{\mathbf{x}}^{(i)}, \mathbf{u}' \rangle \left( \langle \mathbf{u}, \mathbf{u}' \rangle - \frac{1}{2} \langle \tilde{\mathbf{x}}^{(i)}, \mathbf{u} \rangle \langle \tilde{\mathbf{x}}^{(i)}, \mathbf{u}' \rangle - \sum_{j=1}^{i-1} \langle \tilde{\mathbf{x}}^{(j)}, \mathbf{u} \rangle \langle \tilde{\mathbf{x}}^{(j)}, \mathbf{u}' \rangle \right)$$

$$\leq D \sum_{i=1}^{t} \frac{\kappa^2 \tau_i}{D} \left( |\langle \mathbf{u}, \mathbf{u}' \rangle| + \frac{\kappa^2 \tau_i}{2D} + \sum_{j=1}^{i-1} \frac{\kappa^2 \tau_j}{D} \right)$$

$$\leq D \sum_{i=1}^{t} \frac{\kappa^2 \tau_i}{D} \left( |\langle \mathbf{u}, \mathbf{u}' \rangle| + \sum_{j=1}^{i} \frac{\kappa^2 \tau_j}{D} \right)$$

$$= \kappa^2 |\langle \mathbf{u}, \mathbf{u}' \rangle| \sum_{i=1}^{t} \tau_i + \frac{\kappa^4}{D} \left( \sum_{i=1}^{t} \tau_i \right)^2$$

Which completes the proof by substituting this back into Imported Lemma 47:

$$\underset{\mathcal{Z}_t \sim \mathbb{Q}}{\mathbb{E}} \left[ \left( \frac{d\bar{\mathbb{P}}[\mathcal{Z}_t; \mathcal{V}_{\mathbf{u}}^t]}{d\mathbb{Q}(\mathcal{Z}_t)} \right)^2 \right] \leq \underset{\mathbf{u},\mathbf{u}'\sim\mathcal{P}}{\mathbb{E}} \left[ e^{\lambda^2 \kappa^2 |\langle \mathbf{u}, \mathbf{u}' \rangle| \sum_{i=1}^{t} \tau_i + \frac{\lambda^2 \kappa^4}{D} (\sum_{i=1}^{t} \tau_i)^2} \right]$$

$\square$

We can now prove the overall lower bound Theorem 40.

*Proof of Theorem 40.* We apply Imported Lemma 43 to the distribution $\bar{\mathbb{P}}$ truncated to $\mathcal{V}_{\mathbf{u}}^t$. We get that

$$p = \bar{\mathbb{P}}[\mathcal{V}_{\mathbf{u}}^t] = \underset{\mathcal{Z}_t \sim \bar{\mathbb{P}}}{\Pr}[\forall i \in [t] \ : \ \langle \mathbf{v}^{(i)}, \mathbf{u} \rangle^2 \leq \frac{\tau_i}{D}]$$

so that

$$1 - p = \underset{\mathcal{Z}_t \sim \bar{\mathbb{P}}}{\Pr}[\exists i \in [t] \ : \ \langle \mathbf{v}^{(i)}, \mathbf{u} \rangle^2 > \frac{\tau_i}{D}].$$

By Lemma 46, we therefore know that

$$1 - p \leq f(\tau_1) + 2 \sum_{i=1}^{t} e^{\frac{\lambda^2 \kappa^2}{2} \kappa \sum_{j=1}^{i-1} \tau_j} \sqrt{f(\tau_i)}$$

which we are told is at most $z$. We similarly know by Lemma 48 that

$$\underset{\mathcal{Z}_t \sim \mathbb{Q}}{\mathbb{E}} \left[ \left( \frac{d\bar{\mathbb{P}}[\mathcal{Z}_t; \mathcal{V}_{\mathbf{u}}^t]}{d\mathbb{Q}(\mathcal{Z}_t)} \right)^2 \right] \leq \underset{\mathbf{u},\mathbf{u}'\sim\mathcal{P}}{\mathbb{E}} \left[ e^{\lambda^2 \kappa \langle \tilde{\mathbf{u}}, \mathbf{u}' \rangle \sum_{i=1}^{t} \tau_i + \frac{\lambda^2 \kappa^2}{D} (\sum_{i=1}^{t} \tau_i)^2} \right]$$

we are told is at most $1 + z$. So, by Imported Lemma 43, we complete the proof. $\square$

### D.1. Unrolling Lemma

Partially unrelated to the above, we will also need to mildly generalize a technical result from (Simchowitz et al., 2017) that helps handle adaptivity when resolving the adaptive lower bound against L2 estimation. The following is very similar to Lemma C.2 from (Simchowitz et al., 2017):

**Lemma 49** (Unrolling Lemma). *Let $\mathbb{P}_a$, $\mathbb{P}_b$, and $\mathbb{C}$ be distributions over a random variable $\mathcal{Z}_t = (z_1, \cdots, z_t)$ for some arbitrary sample space $z_i \in \Omega$. Let $\mathcal{Z}_i = (z_1, \cdots, z_i)$ for all $i \in [t]$. Let $\{A^i\}_{i \in [t]}$ be a sequence of events such that $A^i$ is deterministic in $\mathcal{Z}_i$ and such that $A^i \subseteq A^{i-1}$. Let $g_i(\mathcal{Z}_{i-1})$ be the expected likelihood ratio between our three distributions at timestep $i$ given $\mathcal{Z}_{i-1}$:*

$$g_i(\mathcal{Z}_{i-1}) = \mathbb{E}\left[\frac{d\mathbb{P}_a(z_i \mid \mathcal{Z}_{i-1})d\mathbb{P}_b(z_i \mid \mathcal{Z}_{i-1})}{(d\mathbb{Q}(z_i \mid \mathcal{Z}_{i-1}))^2}\Big|\mathcal{Z}_{i-1}\right].$$

*Then,*

$$\mathbb{E}\left[\frac{d\mathbb{P}_a(\mathcal{Z}_t)d\mathbb{P}_b(\mathcal{Z}_t)}{(d\mathbb{Q}(\mathcal{Z}_t))^2}\mathbb{1}_{[A^{t-1}]}\right] \le \sup_{\mathcal{Z}_t \in A^{t-1}}\prod_{i=1}^{t}g_i(\mathcal{Z}_{i-1}).$$

*Proof.* We start by defining the tail set $B^i(\mathcal{Z}_i)$ as the set of all $\tilde{z}_{i+1}, \cdots, \tilde{z}_t$ such that $(z_1, \cdots, z_i, \tilde{z}_{i+1}, \cdots, \tilde{z}_t) \in A^t$. We will also define

$$G_i(\mathcal{Z}_i) := \sup_{\tilde{\mathcal{Z}}_t \in B^i(\mathcal{Z}_i)}\prod_{j=i+2}^{t}g_j(\tilde{\mathcal{Z}}_{j-1}).$$

Notice that $G_0(\mathcal{Z}_0) = \sup_{\mathcal{Z}_t \in A^t}\prod_{i=2}^{t}g_i(\mathcal{Z}_{i-1})$, and take $G_{t-1}(\mathcal{Z}_{t-1}) := 1$. Further, notice that for any $\mathcal{Z}_{i-1}$ where the event $A^{i-1}$ holds,

$$G_{i-1}(\mathcal{Z}_{i-1})g_i(\mathcal{Z}_{i-1}) \le \sup_{\tilde{\mathcal{Z}}_t \in B^{i-2}(\mathcal{Z}_{i-2})}G_{i-1}(\tilde{\mathcal{Z}}_{i-1})g_i(\tilde{\mathcal{Z}}_{i-1}) = G_{i-2}(\mathcal{Z}_{i-2}).$$

Then, for any $i \in [t]$, we use tower rule to expand

$$\mathbb{E}\left[\frac{d\mathbb{P}_a(\mathcal{Z}_i)d\mathbb{P}_b(\mathcal{Z}_i)}{(d\mathbb{Q}(\mathcal{Z}_i))^2}\mathbb{1}_{[A^{i-1}]}G_{i-1}(\mathcal{Z}_{i-1})\right]$$

$$= \mathbb{E}\left[\mathbb{E}\left[\frac{d\mathbb{P}_a(\mathcal{Z}_i)d\mathbb{P}_b(\mathcal{Z}_i)}{(d\mathbb{Q}(\mathcal{Z}_i))^2}\mathbb{1}_{[A^{i-1}]}G_{i-1}(\mathcal{Z}_{i-1}) \mid \mathcal{Z}_{i-1}\right]\right]$$

$$= \mathbb{E}\left[\mathbb{E}\left[\frac{d\mathbb{P}_a(\mathcal{Z}_i \mid \mathcal{Z}_{i-1})d\mathbb{P}_b(\mathcal{Z}_i \mid \mathcal{Z}_{i-1})}{(d\mathbb{Q}(\mathcal{Z}_i \mid \mathcal{Z}_{i-1}))^2}\frac{d\mathbb{P}_a(\mathcal{Z}_{i-1})d\mathbb{P}_b(\mathcal{Z}_{i-1})}{(d\mathbb{Q}(\mathcal{Z}_{i-1}))^2}\mathbb{1}_{[A^{i-1}]}G_{i-1}(\mathcal{Z}_{i-1}) \mid \mathcal{Z}_{i-1}\right]\right]$$

$$= \mathbb{E}\left[\frac{d\mathbb{P}_a(\mathcal{Z}_{i-1})d\mathbb{P}_b(\mathcal{Z}_{i-1})}{(d\mathbb{Q}(\mathcal{Z}_{i-1}))^2}\mathbb{1}_{[A^{i-1}]}G_{i-1}(\mathcal{Z}_{i-1})\mathbb{E}\left[\frac{d\mathbb{P}_a(\mathcal{Z}_i \mid \mathcal{Z}_{i-1})d\mathbb{P}_b(\mathcal{Z}_i \mid \mathcal{Z}_{i-1})}{(d\mathbb{Q}(\mathcal{Z}_i \mid \mathcal{Z}_{i-1}))^2} \mid \mathcal{Z}_{i-1}\right]\right]$$

$$= \mathbb{E}\left[\frac{d\mathbb{P}_a(\mathcal{Z}_{i-1})d\mathbb{P}_b(\mathcal{Z}_{i-1})}{(d\mathbb{Q}(\mathcal{Z}_{i-1}))^2}\mathbb{1}_{[A^{i-1}]}G_{i-1}(\mathcal{Z}_{i-1})g_i(\mathcal{Z}_{i-1})\right]$$

$$\le \mathbb{E}\left[\frac{d\mathbb{P}_a(\mathcal{Z}_{i-1})d\mathbb{P}_b(\mathcal{Z}_{i-1})}{(d\mathbb{Q}(\mathcal{Z}_{i-1}))^2}\mathbb{1}_{[A^{i-1}]}G_{i-2}(\mathcal{Z}_{i-2})\right]$$

$$\le \mathbb{E}\left[\frac{d\mathbb{P}_a(\mathcal{Z}_{i-1})d\mathbb{P}_b(\mathcal{Z}_{i-1})}{(d\mathbb{Q}(\mathcal{Z}_{i-1}))^2}\mathbb{1}_{[A^{i-2}]}G_{i-2}(\mathcal{Z}_{i-2})\right]$$

So, by induction, we find that

$$\mathbb{E}\left[\frac{d\mathbb{P}_a(\mathcal{Z}_t)d\mathbb{P}_b(\mathcal{Z}_t)}{(d\mathbb{Q}(\mathcal{Z}_t))^2}\mathbb{1}_{[A^{t-1}]}\right] = \mathbb{E}\left[\frac{d\mathbb{P}_a(\mathcal{Z}_t)d\mathbb{P}_b(\mathcal{Z}_t)}{(d\mathbb{Q}(\mathcal{Z}_t))^2}\mathbb{1}_{[A^{t-1}]}G_{t-1}(\mathcal{Z}_{t-1})\right] \le \mathbb{E}\left[\frac{d\mathbb{P}_a(\mathcal{Z}_1)d\mathbb{P}_b(\mathcal{Z}_1)}{(d\mathbb{Q}(\mathcal{Z}_1))^2}\mathbb{1}_{[A^0]}G_0(\mathcal{Z}_0)\right].$$

We take $A^0$ to be the whole space, so $\mathbb{1}_{[A^0]} = 1$. Also, $G_0(\mathcal{Z}_0) = \sup_{\mathcal{Z}_t \in A^t}\prod_{i=2}^{t}g_i(\mathcal{Z}_{i-1})$. So, we find

$$\mathbb{E}\left[\frac{d\mathbb{P}_a(\mathcal{Z}_t)d\mathbb{P}_b(\mathcal{Z}_t)}{(d\mathbb{Q}(\mathcal{Z}_t))^2}\mathbb{1}_{[A^{t-1}]}\right] \le \left(\sup_{\mathcal{Z}_t \in A^t}\prod_{i=2}^{t}g_i(\mathcal{Z}_{i-1})\right)\mathbb{E}\left[\frac{d\mathbb{P}_a(\mathcal{Z}_1)d\mathbb{P}_b(\mathcal{Z}_1)}{(d\mathbb{Q}(\mathcal{Z}_1))^2}\right] = \left(\sup_{\mathcal{Z}_t \in A^t}\prod_{i=1}^{t}g_i(\mathcal{Z}_{i-1})\right),$$

completing the proof. □

