# OpenReview forum: "Understanding the Kronecker Matrix-Vector Complexity of Linear Algebra"
_ICML.cc/2025/Conference — ICML 2025 poster_

### Official Review · Reviewer_q7YV · 2025-03-10

**Overall Recommendation:** 4

**Summary:**

- The paper addresses several fundamental linear algebraic problems in the Kronecker matrix-vector query model

- Its main contribution is a proof that it requires at least an exponential number of Kronecker matrix-vector products to get a decent estimate of key properties of the matrix

- It also proves that Kronecker matrix-vector algorithms with small-alphabet queries require polynomial complexity for zero testing, whereas it takes O(1) in the non-Kronecker case

- The analysis reveals new insights on the fundamental complexity of linear algebra in the Kronecker matrix-vector model.

**Claims And Evidence:**

All claims made in the submission are well supported by clear and convincing evidence as far as I am aware of.

**Essential References Not Discussed:**

Not that I am aware of.

**Experimental Designs Or Analyses:**

There is no experimental design or analysis in the paper, as it is a pure theoretical work.

**Methods And Evaluation Criteria:**

Proper mathematical tools are used for proving the theoretical claims in the paper.

**Other Comments Or Suggestions:**

- I think there is a typo in Lemma 8 - ... universal constants $C_\tau, C_\delta > 1$ ... should it be $C_\tau, C_0$ instead?

**Other Strengths And Weaknesses:**

- The paper is very well organized and written.

- It is undiscussed how the theoretical merit may affect practical applications that involve Kronecker matrix-vector models

**Questions For Authors:**

How could your result on the exponential complexity of Kronecker matrix-vector models affect its
 applications, such as quantum information science or medical imaging?

**Relation To Broader Scientific Literature:**

- The theoretical result reveals new insights on the fundamental complexity of linear algebra in the Kronecker matrix-vector model.

**Theoretical Claims:**

I have to admit that I did not check the correctness of the proofs due to the time limit.

---

> ### Author Rebuttal · Authors · 2025-04-01
>
> Thanks for the positive feedback, and for catching the typo in Lemma 8 – we will correct it!
>
> * **On the practical impact:** Our exponential ($\exp(q)$) lower bounds for the generic KMVP model serve as a crucial baseline, demonstrating that applications requiring efficiency *cannot* treat the tensor as an arbitrary black-box accessed only via KMVPs ($A(\otimes x_i)$ queries). They *must* exploit known structural properties. This implies two paths forward:
>     1.  **Avoid problematic queries:** Our results on small-alphabet algorithms (Section 5) directly advise against using such sketches (e.g., Rademacher vectors with entries in $\{\pm 1\}$) in downstream applications, as used in practice [Feldman et al., Rakhshan Rabusseau], favoring continuous distributions like Gaussian instead for tasks like zero-testing.
>     2.  **Leverage structure:** Efficient algorithms must use more structural information. Our bounds quantify the "cost of ignorance" when restricted to KMVPs. This justifies why successful tensor methods often use algorithms tailored to specific representations (like [Ahle et al.], exploiting data storage) or leverage query types beyond standard KMVPs that are enabled by that structure (e.g., MPO-vector products for TT-matrices [Rakhshan Rabusseau]).
>     For low-order tensors (e.g., $q=3$ or $q=4$ in medical imaging), an $\exp(q)$ dependence might be tolerable, but our results highlight the scaling limitations for higher-order tensors common in other fields like quantum information science.
>
> Thanks for the question! We will incorporate this discussion into the paper.

---

### Official Review · Reviewer_KWPu · 2025-03-14

**Overall Recommendation:** 4

**Summary:**

This paper studied the Kronecker matrix product oracle complexity lower bounds for estimating the trace and spectrum of a matrix $A$. The authors showed that for a matrix $A\in R^{n^q\times n^q}$ and vector $x = \otimes x_i$, $x_i\in R^d$ and $i\in [q]$, to estimate $tr(A)$ or $\lambda_1(A)$, it requires at least $C^q$ products between $A$ and R$x$. The result relies on a novel probability bound on the inner product of 2 uniform kronecker-product-vectors. The result also implied that if $x_i\in\{1,-1\}^n$, it takes $2^q$ queries to kronecker matrix products to $A$ to test whether $A = 0$ or not, while if $x_i$ are gaussian vectors it only requires 1 query.

**Claims And Evidence:**

Yes

**Essential References Not Discussed:**

N/A

**Experimental Designs Or Analyses:**

Theory paper. N/A

**Methods And Evaluation Criteria:**

Yes

**Other Comments Or Suggestions:**

N/A

**Other Strengths And Weaknesses:**

I think this paper is clearly written with many useful intuitions and explanations. I am not very familiar with the literature on the oracle complexity of tensor calculations using kronecker products. However, I think this paper has a significant contribution in the computational (oracle) complexity for tensor calculations. The paper provided a very interesting separation in the oracle complexity between the methods using non-kronecker products and knronecker products, as well as an separation in the performance of knronecker product based algorithm between large alphabets and small alphabets. The techniques in this paper might also be of independent interest for other fields.

**Questions For Authors:**

I am wondering how would the sparsity of A contributes to the oracle complexity of kronecker products?

**Relation To Broader Scientific Literature:**

The work nices extends the thread of research of the computation complexity of tensor calculations.

**Theoretical Claims:**

I checked the proofs in the main body.

---

> ### Author Rebuttal · Authors · 2025-04-01
>
> We thank the reviewer for their appreciation of our work and the insightful question!
>
> * **On the effect of sparsity:** If $A$ is sparse, we do not particularly expect this to drastically reduce the *oracle* query complexity. Suppose we aim to show that $s$-sparsity reduces complexity. We can take our dense lower bound instance $A \in \mathbb{R}^{n^q \times n^q}$ and embed it into a larger zero-padded matrix $B \in \mathbb{R}^{m^q \times m^q}$ (where $m \approx n/s^{1/q}$) such that $B$ is $s$-sparse but information-theoretically equivalent to $A$ regarding KMVP queries involving the non-zero block. Our $\exp(q)$ lower bounds would still hold for recovering information about the original $A$ via queries to $B$. This indicates that the hardness demonstrated by our lower bounds stems from the limitations of the KMVP queries interacting with worst-case matrix constructions, rather than simply the density of the matrix. It might be possible to get sparsity-dependent bounds with careful parameterization, but simple sparsity alone doesn't bypass the worst-case lower bound in this model.
>
> Thanks for the question!

---

### Official Review · Reviewer_xSH8 · 2025-03-17

**Overall Recommendation:** 2

**Summary:**

Given a matrix $A$, this paper considers estimating the top eigenvalue and the trace of $A$ by matrix-vector multiplications in which the vector is the Kronecker product of $q$ vectors. The main results of the paper are that constant factor estimation of these values require exponential in $q$ such matrix-vector multiplications.

Another result states that testing whether $A=0$ requires exponential in $q$ Kronecker matrix-vector multiplications if the $q$ vectors come from the Radamacher distribution. This is in contrast with the Gaussian case.

All of the results of the paper rely on the fact that a random Kronecker product vector is almost orthogonal to an arbitrary Kronecker product vector with high probability. In fact, the “level of orthogonality” is smaller than what one expects for general vectors by a factor exponential in $q$. This is not very hard to see from the algebraic structure of the Kronecker product.

**Claims And Evidence:**

Some claims regarding the importance of this result for tensor decompositions are not justified in my opinion.

**Essential References Not Discussed:**

I don't know any important references that have not been discussed.

**Experimental Designs Or Analyses:**

The paper is theoretical and this is not applicable.

**Methods And Evaluation Criteria:**

The paper is theoretical and this is not applicable.

**Other Comments Or Suggestions:**

I don't have any other comments or suggestions.

**Other Strengths And Weaknesses:**

The results of the paper are not very surprising in my opinion and they are not aligned with the motivations about tensor decomposition with which the paper starts. More specifically, the paper argues that their results have strong implications for tensor decomposition since in many cases, one does not want to unravel the decomposition’s structure to test properties related to it, and therefore, it is more suitable to consider the matrix-vector multiplication model. However, the results of the paper are for general matrices and not the structured matrices arising from tensor decompositions. I don’t find the results very appealing if they are not generalizable to such structured matrices, and I don’t see how they could be generalized to such matrices.

**Questions For Authors:**

My main question is whether these results are generalizable to the structured matrices that appear in tensor decompositions.

**Relation To Broader Scientific Literature:**

It is not clear what the exact implications of these results are to the broader literature.

Prior works have established exponential bounds for embeddings which I believe are important for the sampling literature, but the current results do not seem to give any implications for that either.

**Theoretical Claims:**

I have not checked the proofs carefully, but the results sound natural and correct to me.

---

> ### Author Rebuttal · Authors · 2025-04-01
>
> We thank the reviewer for raising this critical point about the relationship between our general matrix lower bounds and specific tensor decomposition applications. We understand the concern that our worst-case instances might not be compactly representable. However, our work's primary focus is on understanding the fundamental limitations imposed by the **Kronecker matrix-vector product (KMVP) query model itself**. We view this model, where interaction is limited to queries of the form $A(\otimes x_i)$, as a natural, restricted way to interact with large linear operators (which may or may not arise from tensors), and we aim to characterize its inherent power.
>
> Crucially, as the reviewer notes, some tensors *are* compactly representable. Our results in Section 5 for Zero-Testing provide a concrete example where the lower bound instance *is* a sparse rank-one tensor, exactly representable in standard formats (CP, Tucker, TT, etc.). For this directly relevant case, we prove an exponential ($\exp(q)$) separation between small-alphabet sketches (like Rademacher, used in practice) and large-alphabet sketches (Gaussian), showing the former fail dramatically even for this simple task. This yields the direct, practical implication that small-alphabet sketches should be avoided in tensor algorithms relying on KMVP-like queries.
>
> Regarding the bounds in Section 4 for potentially non-compact matrices: these results are vital because they establish the **baseline difficulty** of solving fundamental problems when restricted *solely* to KMVP queries. The exponential ($\exp(q)$) complexity proves that **any algorithm achieving sub-exponential performance in this model *must* implicitly or explicitly leverage structural properties of the matrix beyond what is accessible through generic KMVP queries.** If an algorithm treats the matrix purely as a black-box accessible only via KMVPs, it *will* fail on our hard instances. Therefore, our work rigorously demonstrates *why* successful tensor algorithms often cannot afford to be structure-oblivious; they must exploit the specific tensor representation (like TT/Tucker) either through specialized queries or direct manipulation of factors. This directly addresses the dichotomy noted in our introduction [Lines 36-41]: our lower bounds explain the inefficiency of structure-oblivious methods operating within the KMVP framework and quantify the inherent cost associated with this restricted oracle access.
>
> In summary, our results provide fundamental insights into the KMVP query model. The zero-testing bounds offer direct practical guidance, while the general lower bounds rigorously justify the necessity of structure-aware algorithm design for tensors when seeking efficient solutions. We hope this clarifies the significance of our findings, and we will refine the paper's exposition to better emphasize these connections. Thank you again for the feedback!

---

### Official Review · Reviewer_VCfy · 2025-03-17

**Overall Recommendation:** 4

**Summary:**

The authors study a computational model where a matrix A can only be accessed through matrix-vector products Ax where x has the specific form of the Kronecker product of q vectors. The paper establishes several key results:
- The authors prove exponential lower bounds (in terms of q) on the number of queries needed to estimate properties such as the trace and top eigenvalue of a matrix, when restricted to using Kronecker matrix-vector products. These bounds apply to all adaptive algorithms under a mild conditioning assumption.
- They demonstrate a fundamental gap between different types of random vectors.
- The core technical insight driving these results is that random vectors with Kronecker structure have exponentially smaller inner products compared to their non-Kronecker counterparts, fundamentally limiting information extraction per query.

**Claims And Evidence:**

The claims made in the paper are supported by rigorous mathematical proofs. The authors provide formal theorems, lemmas, and detailed proofs for their main results. The conditioning assumption they make for their lower bounds (requiring that the matrix of query vectors is not ill-conditioned) is reasonable and well-justified, as they explain why most practical algorithms naturally satisfy this condition. The paper effectively connects theoretical results to practical implications, explaining previously observed phenomena in prior work through a coherent mathematical framework, which adds credibility to their claims.

**Essential References Not Discussed:**

To the best of my understanding, the authors have covered the main results in the literature.

**Experimental Designs Or Analyses:**

This is a theoretical paper without experimental components.

**Methods And Evaluation Criteria:**

The methods used are appropriate. The authors:
- Define the relevant computational model precisely (Kronecker Matrix-Vector Oracle).
- Establish formal proof frameworks for lower bounds.
- Use appropriate information-theoretic techniques adapted from prior work.
- Derive matching upper and lower bounds for certain problems.

**Other Comments Or Suggestions:**

None.

**Other Strengths And Weaknesses:**

Strengths and weaknesses have been highlighted in other questions.

**Questions For Authors:**

1. The results establish exponential lower bounds for the worst case. Are there natural subclasses of matrices or tensors for which the Kronecker matrix-vector complexity is not exponential?
2. How do your results extend to approximate tensor representations? Do the same exponential lower bounds apply when we only seek approximate solutions?
3. How does the trade-off between query complexity and accuracy behave? Specifically, if we relax the accuracy requirements (e.g., allow larger approximation errors), can we achieve sub-exponential query complexity?

**Relation To Broader Scientific Literature:**

The paper effectively connects to several areas of scientific literature, such as lower bounds for matrix-vector algorithms, tensor sketching, Kronecker variants of randomized linear algebra methods, etc.

**Theoretical Claims:**

The theoretical analysis appears sound:
- Lemma 8, which establishes the near-orthogonality of random Kronecker-structured vectors, forms a critical foundation for the other results. The proof appears sound.
- The proofs of Theorems 6 and 7 (lower bounds for spectral norm and trace estimation) build on established information-theoretic techniques but extend them to the Kronecker setting.
- The zero-testing complexity results (Theorem 18) correctly establish both upper and lower bounds for different alphabet sizes.

---

> ### Author Rebuttal · Authors · 2025-04-01
>
> We thank the reviewer for their positive comments and insightful questions!
>
> * **On subclasses with non-exponential complexity:** That's an excellent question exploring the boundary of our worst-case results. Information-theoretically, if a matrix class $\mathcal{A}$ has $D$ degrees of freedom (e.g., $D = O(kqn^2)$ if $A$ is a sum of $k$ Kronecker-structured symmetric matrices), then $\approx D$ KMVP queries should suffice to identify $A \in \mathcal{A}$. However, our work focuses on the *computational* query complexity within the restricted Kronecker matrix-vector product (KMVP) model, where queries are of the form $A(\otimes x_i)$. Our results show that *without* assuming specific structure that is *exploitable* via KMVP queries (or alternative queries enabled by the structure, like Tensor Train-vector products for TT-matrices), the complexity is exponential, specifically $\exp(q)$. Therefore, efficient algorithms *necessitate* leveraging such structural assumptions, confirming that MPO/TT or similar structured classes are indeed the candidates where sub-exponential complexity might be achievable, precisely *because* they allow breaking away from the limitations of the generic KMVP oracle.
>
> * **On approximate tensor representations:** We interpreted this as "given access to an arbitrary tensor $T$, can we show exponential lower bounds against the number of KMVPs needed to recover an approximate compact tensor representation of $T$?” Based on Lemma 45 (appendix), which shows $\exp(q)$ queries are needed to find a Kronecker vector $v$ non-trivially correlated with a planted rank-one Kronecker vector $u$ in noise ($A = W + \lambda uu'$), our answer is partially yes. This suggests that even finding low-rank tensor *structure* within noise is hard in the KMVP model. While we haven't rigorously proven "$\exp(q)$ queries are needed to produce a near-optimal TT approximation to $A$," Lemma 45 strongly indicates the difficulty of recovering even approximate structure efficiently via KMVPs alone.
>
> * **On the accuracy-complexity trade-off:** Our results suggest this trade-off is poor in the KMVP model. For zero-testing from a small alphabet (Section 5), algorithms with polynomial query complexity incur infinite relative error with high probability. For the not-ill-conditioned lower bounds (Section 4), Theorem 6 (spectral norm, $\lambda_1(A)$) already demonstrates exponential multiplicative error even with exponentially many queries. Theorem 7 (trace estimation, $\mathrm{tr}(A)$) likely implies a similar outcome via Lemma 37. Thus, for these fundamental problems, simply relaxing the accuracy requirement to constant multiplicative error does not appear to break the $\exp(q)$ barrier within the KMVP model, according to our analysis.
>
> Thanks again for your questions!

---

> > ### Comment · Reviewer_VCfy · 2025-04-07
> >
> > I would like to thank the authors for their feedback on my questions. I will keep my overall recommendation.

---

### Decision · Program_Chairs · 2025-05-01

**Decision:**

Accept (poster)

**Comment:**

Overall, three of the reviewers found the paper interesting and well-written. However, one of the reviewers raised concerns about the motivation of the Kronecker matrix-vector product model, and thereby, about the significance of the results. These concerns remain after the author response, and should be carefully addressed in the paper.